# ON THE GLOBAL CONVERGENCE OF NATURAL ACTOR-CRITIC WITH NEURAL NETWORK PARAMETRIZATION

## ABSTRACT

Despite the empirical effectiveness of natural actor-critic (NAC) algorithms, their theoretical underpinnings remain relatively unexplored, especially with neural network parameterizations. In the existing literature, the non-asymptotic sample complexity bounds for NAC hold only when the critic is either tabular or are represented by a linear function. In this work, we relax such assumptions for NAC and utilize multi-layer neural network parameterization of the critic and an arbitrary smooth function for the actor. We establish the non-asymptotic sample complexity bounds of $\tilde{\mathcal{O}}\left(\frac{1}{\epsilon^4(1-\gamma)^4}\right)$ for the global convergence of NAC algorithm. We obtain this result using our unique decomposition of the error incurred at each critic step. The critic error is decomposed into the error incurred in fitting the sampled data, the error incurred due to the lack of knowledge of the transition matrix as well as the error incurred due to the limited approximation power of the class of neural networks. In contrast to the existing works for NAC with neural network parameterization of the critic, our analysis does not require i.i.d sampling.

## 1 INTRODUCTION

The use of neural networks in actor-critic (AC) algorithms is widespread in various machine learning applications, such as games (Vinyals et al., 2017; Bonjour et al., 2022), robotics (Morgan et al., 2021), autonomous driving (Kiran et al., 2022), ride-sharing (Al-Abbasi et al., 2019), networking (Geng et al., 2020), and recommender systems (Li et al., 2020). AC algorithms sequentially update the estimate of the actor (policy) and the critic (value function) based on the data collected at each iteration, as described in Konda & Tsitsiklis (1999). An empirical and theoretical improvement of the AC, known as natural actor-critic (NAC), was proposed in Peters & Schaal (2008). NAC replaced the stochastic gradient step of the actor with the natural gradient descent step described in Kakade (2001b) based on the theory of Rattray et al. (1998). The finite-time or non-asymptotic sample complexity bounds for NAC are limited to settings the critic has a linear parametrization (Xu et al., 2020a). However, since linear parametrization is quite restrictive, in practice mostly non-linear neural network-based parameterizations for both actor and critic are used as in Wang et al. (2021a). Despite the widespread use in practice, no finite-time sample complexity bounds are available for the setting when neural networks (NNs) represent the critic in the NAC algorithm.

The linear parametrization allows for a closed-form update for both the actor and the critic update steps. On the other hand, no such closed-form expressions are available for the NAC algorithm with non-linear parametrization. In recent work by Fu et al. (2020), a finite time bound is derived for linear function approximation but for NN parametrization, only an asymptotic convergence is established, moreover, this work requires i.i.d sampling. Wang et al. (2019) establishes similar asymptotic bounds where both the actor and critic are represented using a 2-layer neural network. Hence, we ask this question

*Is it possible to obtain non-asymptotic sample complexity bounds for global convergence of the natural actor-critic algorithm with a multi-layer neural network parametrization of the critic?*

We answer this question by deriving precise non-asymptotic sample complexity bounds for the global convergence of the NAC algorithm. Our approach relies on decomposing the error incurred

Table 1: This table summarizes the sample complexities of different natural actor-critic algorithms. Our result is the first to provide sample complexity results of NAC for a general MDP setting with neural network (NN) parametrization for the critic.

| References | Actor parametrization | Critic parametrization | Sample Complexity |
|---|---|---|---|
| (Xu et al., 2020b) | Linear | Linear | $\tilde{\mathcal{O}}(\epsilon^{-4}(1-\gamma)^{-9})$ |
| (Khodadadian et al., 2021) | Linear | Linear | $\tilde{\mathcal{O}}(\epsilon^{-3}(1-\gamma)^{-11})$ |
| (Xu et al., 2020a) | Linear | Linear | $\tilde{\mathcal{O}}(\epsilon^{-2}(1-\gamma)^{-4})$ |
| (Wang et al., 2019) | 2-layer NN | 2-layer NN | Asymptotic |
| (Fu et al., 2020) | Multi-layer NN | Multi-layer NN | Asymptotic |
| **This work** | Multi-layer NN | Multi-layer NN | $\tilde{\mathcal{O}}(\epsilon^{-4}(1-\gamma)^{-4})$ |

at each step of the NAC algorithm into the errors at the actor and critic steps separately. The main novelty in our approach is that the error incurred in the critic step is decomposed into the error in fitting the observed data, the error incurred due to the lack of knowledge of the transition matrix and the error due to finite approximation power of the class of neural networks. This contrasts the approach in Fu et al. (2020) where both the critic and actor optimizations are analyzed as stochastic gradient descent problems; thus, only an asymptotic error bound is possible with their approach. Hence, we summarize our contributions as follows.

- We derive a non-asymptotic sample complexity bound of $\tilde{\mathcal{O}}(\epsilon^{-4}(1-\gamma)^{-4})$ for the global convergence of the natural actor-critic algorithm with neural network parameterizations for the critic and the actor. To achieve that, our two main novelties in the convergence analysis are highlighted next.

- Building upon the insights presented in Agarwal et al. (2021), we leverage the inherent smoothness property of the actor parametrization to derive an upper bound on the estimation error of the optimal value function. This upper bound is expressed in terms of the error incurred in attaining the compatible function approximation term, as elucidated in Sutton et al. (1999) and the error incurred in estimating the action value function used to solve the compatible function approximation.

- The error incurred at the critic step in fitting the data obtained through sampling is upper bounded using results from Allen-Zhu et al. (2019). The error incurred due to the lack of knowledge of the transition matrix is bounded in terms of the Radamacher complexity of the class of neural networks. This approach allows us to achieve the first non-asymptotic sample complexity bound for NAC with the critic parameterised by a multi layer neural network. It also allows us to have a milder assumption on the error incurred due to the limited approximation the function class representing the critic as compared to other finite time convergence results such as Xu et al. (2020a). Finally, we do not need to assume i.i.d sampling in this approach.

## 2 RELATED WORKS

**Natural Policy Gradient.** The problem of the non-convexity of the critic can be avoided if we use the natural policy gradient algorithm (Kakade, 2001b) where instead of maintaining a parameterized estimate of the critic, we obtain an estimate (or multiple estimates) at each iteration through a Monte Carlo estimate. In such a case, sample complexity estimates are possible without the assumption of linear function approximation of the value function. Agarwal et al. (2021) obtained a sample complexity bound of $\tilde{\mathcal{O}}\left(\frac{1}{\epsilon^4(1-\gamma)^8}\right)$, which was improved to $\tilde{\mathcal{O}}\left(\frac{1}{\epsilon^2(1-\gamma)^7}\right)$ in (Yuan et al., 2022) with the restriction of the actor being represented by a log-linear class of functions. Further improvement was obtained in (Liu et al., 2020b) with a sample complexity of $\tilde{\mathcal{O}}\left(\frac{1}{\epsilon^3(1-\gamma)^6}\right)$ and also did not require the restriction to log-linear class of functions to represent the actor. In spite of obtaining finite time sample complexity bounds, the Natural Policy Gradient algorithms suffer from high variance due to the Monte Carlo estimate. Additionally, each estimate of the critic requires on average a sample of size $\left(\frac{1}{1-\gamma}\right)$, thus these algorithms are not sample efficient in terms of $\gamma$. Additionally the

error incurred due to the Monte Carlo sampling of the critic as well as the lack of expressability of the class of functions representing the policy is represented as a constant.

**Actor-Critic Methods**. First conceptualized in Sutton (1988), aim to combine the benefits of the policy gradient methods and $Q$-learning based methods. The policy gradient step in these methods is replaced by a Natural Policy Gradient proposed in (Kakade, 2001b) to obtain the so-called Natural Actor Critic in (Peters et al., 2005). Sample complexity results for Actor Critic were first obtained for MDP with finite states and actions in (Williams & Baird, 1990), and more recently in (Lan, 2023; Zhang et al., 2020). Finite time convergence for natural actor critic using a linear MDP assumption has been obtained in (Chen & Zhao, 2022; Khodadadian et al., 2021; Xu et al., 2020b) with the best known sample complexity of $\tilde{\mathcal{O}}\left(\frac{1}{\epsilon^2(1-\gamma)^4}\right)$ (Xu et al., 2020a). Finite time sample complexity results are however, not available for Natural Actor Critic setups for general MDP where neural networks are used to represent the critic. (Fu et al., 2020) obtained asymptotic results for a variant of the Natural Actor Critic using a PPO update for the policy gradient step, but it forgoes the use of the 'clipped surrogate objective', which makes the algorithm unsuitable practically. The key related works here are summarized in Table 1.

## 3 PROBLEM FORMULATION

We consider a discounted Markov Decision Process (MDP) given by the tuple $\mathcal{M} := (\mathcal{S}, \mathcal{A}, P, R, \gamma)$, where $\mathcal{S}$ is a bounded measurable state space, $\mathcal{A}$ is the finite set of actions. $P : \mathcal{S} \times \mathcal{A} \to \mathcal{P}(\mathcal{S})$ is the probability transition kernel[1],$R : \mathcal{S} \times \mathcal{A} \to \mathcal{P}([0, R_{\max}])$ is the reward kernel on the state action space with $R_{\max}$ being the absolute value of the maximum reward, and $0 < \gamma < 1$ is the discount factor. A policy $\pi : \mathcal{S} \to \mathcal{P}(\mathcal{A})$ maps a state to a probability distribution over the action space. The action value function for a given policy $\pi$ is given by

$$Q^\pi(s, a) = \mathbb{E}\left[\sum_{t=0}^{\infty} \gamma^t r(s_t, a_t)|s_0 = s, a_0 = a\right], \tag{1}$$

where $r(s_t, a_t) \sim R(\cdot|s_t, a_t)$, $a_t \sim \pi(\cdot|s_t)$ and $s_{t+1} \sim P(\cdot|s_t, a_t)$ for $t = \{0, \cdots, \infty\}$. For a discounted MDP, we define the optimal action value functions as

$$Q^*(s, a) = \sup_\pi Q^\pi(s, a), \qquad \forall (s, a) \in \mathcal{S} \times \mathcal{A}. \tag{2}$$

A policy that achieves the optimal action-value functions is known as the *optimal policy* and is denoted as $\pi^*$. Similarly, we can define the value function as $V^\pi(s) = \mathbb{E}\left[\sum_{t=0}^{\infty} \gamma^t r'(s_t, a_t)|s_0 = s\right]$, and from the definition of $Q^\pi(s, a)$, it holds that $V^\pi(s) = \mathbb{E}_{a\sim\pi}[Q^\pi(s, a)]$. Similarly, we can define the optimal value function as $V^*(s) = \sup_\pi V^\pi(s), \qquad \forall s \in \mathcal{S}$.

We define $\rho_\nu^\pi(s)$ as the stationary state distribution induced by the policy $\pi$ starting at state distribution $\nu$ and $\zeta_\nu^\pi(s, a)$ is the corresponding stationary state action distribution defined as $\zeta_\nu^\pi(s, a) = \rho_\nu^\pi(s)\pi(a|s)$. We further define $V^\pi(\nu) = \mathbb{E}_{s_0\sim\nu}[V^\pi(s_0)]$, where $\nu$ is an initial state distribution. We can define the visitation distribution as $d_{s_0}^\pi(s) = (1-\gamma)\sum_{t=0}^{\infty} \gamma^t Pr^\pi(s_t = s|s_o)$. Here $Pr^\pi(s_t = s|s_o)$ denotes the probability the state at time $t$ is $s$ given a starting state of $s_o$. Hence, we can write $d_\rho^\pi(s) = \mathbb{E}_{s_o\sim\rho}[d_{s_0}^\pi(s)]$. Finally for any measurable function $f : \mathcal{S} \times \mathcal{A} \to \mathbb{R}$ and a measure $\nu$ defined on $\mathcal{S} \times \mathcal{A}$ we define $\mathbb{E}(f)_\nu = \int_{\mathcal{S}\times\mathcal{A}} f d\nu,$.

We additionally define the bellman operator for a policy $\pi$ on a function $Q : \mathcal{S} \times \mathcal{A} \to \mathbb{R}$ is defined as

$$(T^\pi Q)(s, a) = \mathbb{E}(r(s, a)) + \gamma \int Q(s', \pi(s'))P(ds'|s, a) \tag{3}$$

Further, operator $P^\pi$ is defined as

$$P^\pi Q(s, a) = \mathbb{E}[Q(s', a')|s' \sim P(\cdot|s, a), a' \sim \pi(\cdot|s')] \tag{4}$$

This is the one step Markov transition operator for policy $\pi$ for the Markov chain defined on $\mathcal{S} \times \mathcal{A}$ with the transition dynamics given by $S_{t+1} \sim P(\cdot|S_t, A_t)$ and $A_{t+1} \sim \pi(\cdot|S_{t+1})$. It defines a distribution on the state action space after one transition from the initial state. Similarly, $P^{\pi_1}P^{\pi_2}\cdots P^{\pi_m}$ is the $m$-step Markov transition operator following policy $\pi_t$ at steps $1 \le t \le m$.

---

[1]For a measurable set $\mathcal{X}$, let $\mathcal{P}(\mathcal{X})$ denote the set of all probability measures over $\mathcal{X}$.

## 4 Natural Actor Critic Algorithm Overview

We now describe our natural actor-critic (NAC) algorithm. In a natural policy gradient algorithm (Kakade, 2001a), the policy is parameterized as $\{\pi_\lambda, \lambda \in \Lambda\}$ and $\Lambda \subset \mathbb{R}^d$ where $d$ is a positive integer. We have $K$ total iterations of the Algorithm. At iteration $k$, the policy parameters are updated using a natural policy gradient step given by

$$\lambda_{k+1} = \lambda_k + \eta F_\nu^\dagger(\lambda) \nabla_\lambda V^{\pi_\lambda}(\nu), \tag{5}$$

From the policy gradient theorem in (Sutton et al., 1999) we have

$$\nabla_{\lambda_k} V^{\pi_{\lambda_k}}(\nu) = \mathbb{E}_{s,a}(\nabla \log(\pi_{\lambda_k})(a|s) Q^{\pi_{\lambda_k}}(s,a)), \tag{6}$$

$$F_\nu(\lambda_k) = \mathbb{E}_{s,a}\left[\nabla \log \pi_{\lambda_k}(a|s) \left(\nabla_t \log \pi_{\lambda_k}(a|s)\right)^\top\right], \tag{7}$$

where $s \sim d_\nu^{\pi_{\lambda_k}}, a \sim \pi_{\lambda_k}(.|s)$. From Sutton et al. (1999), the principle of compatible function approximation implies that we have

$$F_\nu^\dagger(\lambda_k) \nabla_{\lambda_k} V^{\pi_{\lambda_k}}(\nu) = \frac{1}{1-\gamma} w_k^* \tag{8}$$

$$w_k^* = \arg\min_w \mathbb{E}_{s,a}(A^{\pi_{\lambda_k}}(s,a) - w \nabla_\lambda \log(\pi_{\lambda_k}(a|s)))^2, \tag{9}$$

and $s \sim d_\nu^{\pi_{\lambda_k}}, a \sim \pi_{\lambda_k}(.|s)$ Here $(A^{\pi_{\lambda_k}}(s,a) = Q^{\pi_{\lambda_k}}(s,a) - V^{\pi_{\lambda_k}}(s))$ and where $F^\dagger$ denotes the Moore-Penrose pseudo-inverse of the matrix $F$. For natural policy gradient algorithms such as in Agarwal et al. (2021) and Liu et al. (2020b) an estimate of $Q^{\pi_{\lambda_k}}$ (and from that an estimate of $A^{\pi_{\lambda_k}}(s,a)$) is obtained through a sampling procedure that requires on average $\left(\frac{1}{1-\gamma}\right)$ for each sample of $Q^{\pi_{\lambda_k}}$ (and thus $A^{\pi_{\lambda_k}}$). For the natural actor-critic setup, we maintain a parameterized estimate of the $Q$-function, which is updated at each step and is used to approximate $Q^{\pi_{\lambda_k}}$. In our case, a neural network with $L$ layers and at least $m$ neurons per layer is used to represent the $Q$ function, at each iteration $k$ of the algorithm, an estimate of its parameters is obtained by solving an optimization of the form

$$\arg\min_{\theta \in \Theta} \mathbb{E}_{s,a}(Q^{\pi_{\lambda_k}} - Q_\theta)^2, \tag{10}$$

Where $(s,a) \sim \zeta_\nu^{\pi_{\lambda_k}}$, $\Theta$ is the space of parameters for the neural networks and $Q_\theta$ is the neural network corresponding to the parameter $\theta$. This step is known as the critic step. A DQN like algorithm to get an estimate of $Q^{\pi_{\lambda_k}}$, as is done in practical implementations of the Natural Actor Critic like Wang et al. (2021a). We summarize the Natural Actor-Critic approach in Algorithm 1. It has one main for loop indexed by the iteration counter $k$. The first inner for loop indexed by $j$ is the loop where the critic step is performed. At a fixed iteration $k$ of the main for loop and iteration $j$ of the first inner for loop, we solve the following optimization problem

$$\arg\min_{\theta \in \Theta} \mathbb{E}_{s,a}(T^{\pi_{\lambda_k}} Q_{k,j-1}(s,a) - Q_\theta(s,a))^2, \tag{11}$$

This is equivalent to the *target network* feature of the Deep Q Network(DQN) algorithm. For the inner loop at iteration $j$, the target is fixed to be $T^{\pi_{\lambda_k}} Q_{k,j-1}(s,a)$. The first inner for loop has a nested inner for loop indexed by $i$ where the optimization step for the current target is performed. The target network is updated at the end of the first inner loop. We note that the *target network* technique is applied in most real-world applications of natural actor critic with neural network critic as in (Wei et al., 2019). The first inner loop controls how many times the target network is updated. To get rid of the Markov dependence between the samples, the *replay buffer* technique is used wherein we randomly sample from the collected data instead of using it sequentially. For the sake of generality, we have not used this as our analysis will account for the Markov dependence between the samples.

The estimate of $w_k^*$ is obtained in the second inner for loop of Algorithm equation 1 indexed by $i$ where a gradient descent is performed for the loss function of the form given in equation 9 using the state action pairs sampled in the first inner for loop. Note that we do not have access to the true advantage function required for the critic update. Thus, we use the estimate of the $Q$ function obtained at the end of the first inner for loop to calculate the advantage function. After obtaining our estimate of the minimizer of equation 9, we update the policy parameter using the stochastic gradient update step. Here, the state action pairs used are the same we sampled in the first inner for loop.

---

**Algorithm 1** Natural Actor Critic with Neural Parametrization

---

**Input:** $\mathcal{S}$, $\mathcal{A}$, $\gamma$, Time Horizon K $\in$ $\mathcal{Z}$ , Updates per time step J $\in$ $\mathcal{Z}$ ,starting state sampling distribution $\nu$, Actor step sizes $\beta_{i,k}, \forall k \in \{1, \cdots, K\}, i \in \{1, \cdots, n.J\}$, Critic step size $\alpha$, policy gradient step size $\eta$,

1: **Initialize:** $\lambda_0 = \{0\}^d$,
2: **for** $k \in \{1, \cdots, K\}$ **do**
3:      Initialize $X = \varnothing, Q_k(s, a) = 0$     $\forall(s, a) \in \mathcal{S} \times \mathcal{A}$
4:      **for** $j \in \{1, \cdots, J\}$ **do**
5:          Sample $s_1$ from $\nu$ and $a_1$ by following $\pi_{\lambda_k}$
6:          Initialize $\theta_0$ using a standard Gaussian.
7:          **for** $i \in \{1, \cdots, n\}$ **do**
8:             Sample the tuple $s_{i+1}, a_{i+1}$ by following the policy $\pi_{\lambda_k}$
9:             Set $y_i = r^{'}(s_i, a_i) + \gamma Q_k(s_{i+1}, a_{i+1})$,
10:             $\theta_i = \theta_{i-1} + \alpha(y_i - Q_{\theta_i}(s_i, a_i))\nabla Q_{\theta_i}(s_i, a_i)$
11:          **end for**
12:          $Q_k = Q_{\theta_n}$
13:          Append the $n$ $(s_i, a_i)$ pairs to the data-set $X$
14:      **end for**
15:      Initialize $w_0 = 0^d$
16:      **for** $i \in \{1, \cdots, |X|\}$ **do**
17:          $A_k(s_i, a_i) = Q_k(s_i, a_i) - \sum_{a \in \mathcal{A}} \pi_{\lambda_k}(a|s_i)Q_k(s_i, a)$
18:          $w_i = w_{i-1} - \beta_{i,k}\left(w_i \cdot \nabla_\lambda \log \pi_{\lambda_k}(a_i|s_i) - A_k(s_i, a_i)\right)\nabla_\lambda \log \pi_{\lambda_k}(a_i|s_i)$
19:      **end for**
20:      Update $\lambda_{k+1} = \lambda_k + \eta w_{|X|}$
21: **end for**
     Output: $\pi_{\lambda_{K+1}}$

---

## 5 GLOBAL CONVERGENCE RESULT

### 5.1 ASSUMPTIONS

Before stating the main result, we formally describe the required assumptions in this subsection.

**Assumption 1.** *For any $\lambda_1, \lambda_2 \in \Lambda$ and $(s, a) \in (\mathcal{S} \times \mathcal{A})$ we have*

$$\|\nabla log(\pi_{\lambda_1})(a|s) - \nabla log(\pi_{\lambda_2})(a|s)\|_2 \le \beta\|\lambda_1 - \lambda_2\|_2 \tag{12}$$

*where $\beta > 0$.*

Such assumptions have been utilized in prior policy gradient based works such as Agarwal et al. (2021); Liu et al. (2020b) and finite time analysis of NAC using linear critic such as Xu et al. (2020a). This assumption is satisfied for the softmax policy parameterization

$$\pi_\lambda(a|s) = \frac{\exp(f_\lambda(s, a))}{\sum_{a' \in \mathcal{A}} \exp(f_\lambda(s, a'))} \tag{13}$$

where $f_\lambda(s, a)$ is a neural network with a smooth activation function. This is the most common form of the policy used in practice (Wei et al., 2019; Wang et al., 2021a). This assumption is also satisfied for Gaussian (Doya, 2000) and Boltzmann policies (Konda & Tsitsiklis, 1999). Thus our analysis is more general than Fu et al. (2020) which is restricted to energy-based policies.

**Assumption 2.** *For any $\lambda \in \Lambda$, let $\pi_\lambda$ be the corresponding policy, $\nu$ be the starting distribution over the state space, and let $\zeta_\nu^{\pi_\lambda}$ be the corresponding stationary state action distribution. We assume that there exists a positive integer $p$ such that for every positive integer $\tau$*

$$d_{TV}\left(\mathbb{P}((s_\tau, a_\tau) \in \cdot|(s_0, a_0) = (s, a)), \zeta_\nu^{\pi_\lambda}(\cdot)\right) \le p\rho^\tau, \forall(s, a) \in \mathcal{S} \times \mathcal{A} \tag{14}$$

This assumption implies that the Markov chain is geometrically mixing. Such assumption is widely used both in the analysis of stochastic gradient descent literature such as Doan (2022); Sun et al.

(2018), as well as finite time analysis of RL algorithms such as Xu et al. (2020a). In Fu et al. (2020), it is assumed that data can be sampled from the stationary distribution of a given policy. We note that this is not possible in practice. Instead, we can only sample from a Markov chain which has a stationary distribution as the desired distribution to sample from.

**Assumption 3.** *For any fixed $\lambda \in \Lambda$ and $\theta \in \Theta$ we have*

$$\min_w \mathbb{E}_{s,a\sim\zeta_\nu^{\pi_{\lambda_k}}} \left(A_\theta(s,a) - w^\top \nabla \log(\pi_\lambda)(a|s)\right)^2 \le \epsilon_{bias} \tag{15}$$

Similar assumptions are made in Fu et al. (2020), where this error is assumed to be zero when the critic has a linear function parameterization. In policy gradient works such as Liu et al. (2020b), the assumption replaces the parameterised estimate of the advantage function $A_\theta$ (which is known to us) with the true advantage function for policy $\pi_\lambda$ denoted by $A^{\pi_\lambda}$ (which is unknown to us). Doing so ignores the error that is incurred due to a mismatch in the actor and critic parameterization which is a critical aspect of a successful implementation of natural actor-critic algorithms. In Xu et al. (2020a), this assumption is implicit as this term is defined as a constant denoted by $\zeta_{approx}^{actor}$.

**Assumption 4.** *For any fixed $\theta \in \Theta$ and $\lambda \in \Lambda$ we have*

$$\min_{\theta_1 \in \Theta} \mathbb{E}_{s,a\sim\zeta_\nu^{\pi_{\lambda_k}}} \left(Q_{\theta_1}(s,a) - T^{\pi_\lambda}Q_\theta(s,a)\right)^2 \le \epsilon_{approx} \tag{16}$$

This assumption is key to the validity of the DQN step. Note that in works such as Xu et al. (2020a), an upper bound is placed on the approximation error when the function class (in that case linear functions) are used to approximate the unknown true value function (see term denoted as $\zeta_{critic}^{approx}$). Our assumption is weaker as we only require the class of neural network to be able to approximate the function obtained by applying the bellman operator to a neural network belonging to the same class.

## 5.2 MAIN RESULT

**Theorem 1.** *Suppose Assumptions 1-4 hold and we have $\alpha = \Theta\left(\frac{1}{poly(n,L).m}\right)$, $\beta_{i,k} = \frac{2}{\mu_k(i+1)}$ where $\mu_k$ is the strong convexity parameter of the loss function in equation 9, $\eta = \frac{1}{\sqrt{K}}$ and $m \ge \mathcal{O}(K.J.\delta^{-1})$ then from Algorithm 1 we obtain with probability at least $1 - \delta$*

$$\min_{k \le K}(V^*(\nu) - V^{\pi_{\lambda_k}}(\nu)) \le \mathcal{O}\left(\frac{1}{\sqrt{K}(1-\gamma)}\right) + \frac{1}{K(1-\gamma)}\sum_{k=1}^{K}\left(\mathcal{O}\left(\frac{\log(J \cdot n)}{J \cdot n}\right) + \mathcal{O}(\gamma^J)\right)$$

$$+ \frac{1}{K(1-\gamma)}\sum_{k=1}^{K}\sum_{j=0}^{J-1}\left(\mathcal{O}\left(1 - \Omega\left(\frac{\alpha m}{n^2}\right)\right)^n + \mathcal{O}\left(\frac{1}{\sqrt{n}}\right)\right)$$

$$+ \frac{1}{1-\gamma}\left(\mathcal{O}(\epsilon_{bias}) + \mathcal{O}(\sqrt{\epsilon_{approx}})\right). \tag{17}$$

*Hence, for $K = \mathcal{O}(\epsilon^{-2}(1-\gamma)^{-2})$, $J = \mathcal{O}\left(\log\left(\frac{1}{\epsilon}\right)\right)$, $n = \tilde{\mathcal{O}}\left(\epsilon^{-2}(1-\gamma)^{-2}\right)$, $m \ge \mathcal{O}(\epsilon^{-2}.\delta^{-1})$*

$$\min_{k \le K}(V^*(\nu) - V^{\pi_{\lambda_k}}(\nu)) \le \epsilon + \frac{1}{1-\gamma}\left(\epsilon_{bias} + (\sqrt{\epsilon_{approx}})\right), \tag{18}$$

*which implies a sample complexity of $K \cdot J \cdot n = \tilde{\mathcal{O}}\left(\epsilon^{-4}(1-\gamma)^{-4}\right)$.*

**Remark 1:** We note that there are seven terms on the right-hand side of equation 17. The first term is a consequence of the smoothness property of the actor parameterization. The second term is the error incurred in estimating $w_k^*$. The third term is the error incurred due to the inherent randomness of the system during each critic update step, in Farahmand et al. (2010) this was known as the *statistical error*. The fourth term on the right is the error incurred in fitting the data at each fixed target in the critic step. The fifth term on the right is the error incurred due to a lack of knowledge of the transition matrix. The sixth term $\epsilon_{bias}$ represents the minimum possible attainable value of the loss function in the actor step. It is also a measure of how *compatible* are the architecture of the actor and critic. In Wang et al. (2019) it is shown that for an over-parameterized neural network used to represent both

the actor and critic this error is zero. The term $\epsilon_{approx}$ is a measure of how well the class of neural networks we use to represent the critic can approximate a function obtained by applying the bellman operator to a function from that same class. Works such as Fan et al. (2020); Chen & Jiang (2019) set this error to zero. The requirement on the minimum number of neurons $m$ in each layer of the critic network can be be thought of as a consequence of the *universal approximation* property which states that sufficiently wide neural networks(even those with a single hidden layer) can approximate any continuous function with arbitrary accuracy.

**Remark 2:** Our sample complexity when compared to the existing state of the art sample complexity bound for natural policy gradient with non-linear policy parameterization of $\tilde{\mathcal{O}}\left(\epsilon^{-3}(1-\gamma)^{-6}\right)$ achieved in Liu et al. (2020a) reveals a key insight. Note that our bound is worse off in terms of $\epsilon$ by a factor of $\epsilon^{-1}$. This is due to the fact that we have to obtain an estimate of the critic parameters while the natural policy gradient does not. We can see this in our result from the fifth term on the right hand side of equation 17 which is $\mathcal{O}(n^{-\frac{1}{2}})$ which is from the critic optimization step. The natural policy gradient algorithm requires on average $(1-\gamma)^{-1}$ state action samples for every sample of $Q(s,a)$. This is reflected in our results as our error bounds are better in terms of $(1-\gamma)$ by a factor of $(1-\gamma)^{-2}$. We discuss this detail in Appendix E.

**Remark 3:** Note the presence of the probability term for our convergence result. This term is present due to the fact that the optimization for the critic step is non-convex, hence convergence can only be guaranteed with a high probability. We show in the Appendix F that if the critic is represented by a two layer neural network with ReLU activation, using the convex reformulation as laid out in Mishkin et al. (2022), a deterministic upper bound on the error can be obtained.

# 6 PROOF SKETCH OF THEOREM 1

The proof is split into two stages. In the first stage, we demonstrate how the difference in value functions is upper bounded as a function of the errors incurred till the final step $K$. The second part is to upper bound the different error components.

**Upper Bounding Error in Separate Error Components:** We use the smoothness property assumed in Assumption 1 to obtain a bound on the expectation of the difference between our estimated value function and the optimal value function.

$$\min_{k \in \{1, \cdots, K\}} V^*(\nu) - V^{\pi_{\lambda_K}}(\nu) \leq \frac{\log(|\mathcal{A}|)}{K\eta(1-\gamma)} + \frac{\eta \beta W^2}{2(1-\gamma)} + \frac{1}{K}\sum_{k=1}^{K} \frac{err_k}{1-\gamma}, \qquad (19)$$

where

$$err_k = \mathbb{E}_{s \sim d_\nu^{\pi^*}, a \sim \pi^*(.|s)}(|A^{\pi_{\lambda_k}} - w^k(s,a)\nabla log(\pi_{\lambda_k}(a|s))|), \qquad (20)$$

where $W$ is a constant such that $||w^k||_2 \leq W \ \forall k$, where $k$ denotes the iteration of the outer for loop of Algorithm 1. We split the term in equation 20 into the errors incurred due to the actor and critic step as follows

$$err_k = \mathbb{E}_{s,a}(|A^{\pi_{\lambda_k}} - w^k \nabla log(\pi_{\lambda_k}(a|s))|) \qquad (21)$$

$$\leq \underbrace{\mathbb{E}_{s,a}(|A^{\pi_{\lambda_k}} - A_{k,J}|)}_{I} + \underbrace{\mathbb{E}_{s,a}(|A_{k,J} - w^k \nabla log(\pi_{\lambda_k}(a|s))|)}_{II}. \qquad (22)$$

Note that $I$ is the difference between the true $A^{\pi_{\lambda_k}}$ function corresponding to the policy $\pi_{\lambda_k}$ and $A_{k,J}$ is our estimate. This estimation is carried out in the first inner for loop of Algorithm 1. Thus $I$ is the error incurred in the critic step. $II$ is the error incurred in the estimation of the actor update. This is incurred in the stochastic gradient descent steps in the second inner for loop of Algorithm 1.

**Upper Bounding Error in Critic Step:** For each iteration $k$ of the Algorithm 1. We show that minimizing $I$ is equivalent to solving the following problem

$$\arg\min_{\theta \in \Theta} \mathbb{E}_{s,a}(Q^{\pi_{\lambda_k}} - Q_\theta)^2, \qquad (23)$$

where $(s, a) \sim \zeta_\nu^{\pi_{\lambda_k}}$. We recreate the result for the value function from Lemmas 2 of Munos (2003) for the action value function $Q$ to obtain

$$\mathbb{E}_{s,a}|Q^{\pi_{\lambda_k}} - Q_{k,J}| \quad \leq \quad \sum_{j=1}^{J-1} \gamma^{J-j-1}(P^{\pi_{\lambda_k}})^{J-j-1}\mathbb{E}|\epsilon_{k,j}| + \gamma^J \left( \frac{R_{max}}{1-\gamma} \right), \tag{24}$$

where $\epsilon = T^{\pi_{\lambda_k}} Q_{k,j-1} - Q_{k,j}$ is the Bellman error incurred at iteration $j$ of the first inner for loop and iteration $k$ of the outer for loop of Algorithm 1. The first term on the right hand side is called as the algorithmic error, which depends on how good our approximation of the Bellman error is. The second term on the right hand side is called as the statistical error, which is the error incurred due to the random nature of the system. Intuitively, the Bellman error depends on how much data is collected at each iteration, how efficient our solution to the optimization step is to the true solution, and how well our function class can approximate $T^{\pi_{\lambda_k}} Q_{k,j-1}$. Building upon this intuition, we split $\epsilon$ into four different components as follows.

$$\begin{aligned} \epsilon_{k,j} &= T^{\pi_{\lambda_k}} Q_{k,j-1} - Q_{k,j} \\ &= \underbrace{T^{\pi_{\lambda_k}} Q_{k,j-1} - Q_{k,j}^1}_{\epsilon_{k,j}^1} + \underbrace{Q_{k,j}^1 - Q_{k,j}^2}_{\epsilon_{k,j}^2} + \underbrace{Q_{k,j}^2 - Q_{k,j}^3}_{\epsilon_{k,j}^3} + \underbrace{Q_{k,j}^3 - Q_{k,j}}_{\epsilon_{k,j}^4} \\ &= \epsilon_{k,j}^1 + \epsilon_{k,j}^2 + \epsilon_{k,j}^3 + \epsilon_{k,j}^4, \end{aligned} \tag{25}$$

We now define the terms introduced above. We first define the various $Q$-functions which we can approximate in decreasing order of the accuracy and then define the corresponding errors.

We start by defining the best possible approximation of the function $T^{\pi_{\lambda_k}} Q_{k,j-1}$ possible from the class of neural networks with smooth activation functions, with respect to the expected square from the true ground truth $T^{\pi_{\lambda_k}} Q_{k,j-1}$.

**Definition 1.** *For iteration $k$ of the outer for loop and iteration $j$ of the first inner for loop of Algorithm 1, we define*

$$Q_{k,j}^1 = \underset{Q_\theta, \theta \in \Theta}{\arg\min} \, \mathbb{E}(Q_\theta(s, a) - T^{\pi_{\lambda_k}} Q_{k,j-1}(s, a))^2, \tag{26}$$

*where $(s, a) \sim \zeta_\nu^{\pi_{\lambda_k}}(s, a)$.*

Note that we do not have access to the transition probability kernel $P$, hence we do not know $T^{\pi_{\lambda_k}}$. To alleviate this, we use the observed next state and actions instead. Using this, we define $Q_{k,j}^2$ as,

**Definition 2.** *For iteration $k$ of the outer for loop and iteration $j$ of the first inner for loop of Algorithm 1, we define*

$$Q_{k,j}^2 = \underset{Q_\theta, \theta \in \Theta}{\arg\min} \, \mathbb{E}(Q_\theta(s, a) - (r'(s, a) + \gamma Q_{k,j-1}(s', a')))^2, \tag{27}$$

*where $(s, a) \sim \zeta_\nu^{\pi_{\lambda_k}}(s, a), s' \sim P(s'|s, a), r'(\cdot|s, a) \sim R(\cdot|s, a)$ and $a' \sim \pi_{\lambda_k}(.|s')$*

To obtain $Q_{k,j}^2$, we still need to compute the true expected value in Equation 27. However, we still do not know the transition function $P$. To remove this limitation, we use sampling. Consider the set of $n$ state-action pairs sampled by starting from a state action distribution $\nu$ and following policy $\pi^{\lambda_k}$, using which we define $Q_{k,j}^3$ as,

**Definition 3.** *For the set of $n$ state action pairs sampled in iteration $k$ of the outer for loop and iteration $j$ of the first inner for loop of Algorithm 1 we define*

$$Q_{k,j}^3 = \underset{Q_\theta, \theta \in \Theta}{\arg\min} \, \frac{1}{n} \sum_{i=1}^{n} \left( Q_\theta(s_i, a_i) - \left( r(s_i, a_i) + \gamma Q_{k,j-1}(s_{i+1}, a_{i+1}) \right) \right)^2, \tag{28}$$

$Q_{k,j}^3$ is the best possible approximation for $Q$-value function which minimizes the sample average of the square loss functions with the target values as $\left( r'(s_i, a_i) + \gamma Q_{k,j-1}(s_{i+1}, a_{i+1}) \right)$. In other words this is the optimal solution for fitting the observed data.

We now defined the errors using the $Q$ functions just defined. We start by defining the approximation error which represents the difference between the function $T^{\pi_{\lambda_k}} Q_{j-1}$ and its best approximation possible from the class of neural networks used for critic parametrization denoted by $Q_{k,j}^1$.

**Definition 4** (Approximation Error). *For a given iteration $k$ of the outer for loop and iteration $j$ of the first inner for loop of Algorithm 1, we define, $\epsilon_{k,j}^1 = T^{\pi_{\lambda_k}} Q_{k,j-1} - Q_{k,j}^1$, where $Q_{k,j-1}$ is the estimate of the $Q$ function at iteration $k$ of the outer for loop and iteration $j-1$ of the first inner for loop of Algorithm 1.*

This error is a measure of the approximation power of the class of neural networks we use to represent the critic. We upper bound this error in lemma 3 in Appendix B.

We also define Estimation Error which denotes the error between the best approximation of $T^{\pi_{\lambda_k}} Q_{k,j-1}$ possible from the class of neural networks denoted by $Q_{k,j}^1$ and the minimizer of the loss function in equation 27 denoted $Q_{k,j}^2$.

**Definition 5** (Estimation Error). *For a given iteration $k$ of the outer for loop and iteration $j$ of the first inner for loop of Algorithm 1, we define, $\epsilon_{k,j}^2 = Q_{k,j}^1 - Q_{k,j}^2$.*

We demonstrate that this error is zero in lemma 4 in Appendix B.

We now define Sampling error which denotes the difference between the minimizer of expected loss function in equation 27 denoted by $Q_{k,j}^2$ and the minimizer of the empirical loss function in equation 28 denoted by $Q_{k,j}^3$. We can see that intuitively, the more samples we have the closer these two functions will be. We use Rademacher complexity results to upper bound this error.

**Definition 6** (Sampling Error). *For a given iteration $k$ of the outer for loop and iteration $j$ of the first inner for loop of Algorithm 1, we define, $\epsilon_{k,j}^3 = Q_{k,j}^3 - Q_{k,j}^2$.*

An upper bound on this error is established in 5 in Appendix B.

Lastly, we define optimization error which denotes the difference between the minimizer of the empirical square loss function, $Q_{k_3}$, and our estimate of this minimizer that is obtained from the gradient descent algorithm.

**Definition 7** (Optimization Error). *For a given iteration $k$ of the outer for loop and iteration $j$ of the first inner for loop of Algorithm 1, we define, $\epsilon_k^4 = Q_{k,j}^3 - Q_{k,j}$. Here $Q_{k,j}$ is our estimate of the $Q$ function at iteration $k$ of Algorithm 1 and iteration $j$ of the first inner loop of Algorithm 1.*

The upper bound on these error terms is established in lemma 6 in Appendix B.

**Upper Bounding Error in Actor Step:** Note that we require the minimization of the term $\mathbb{E}_{s,a}(A_{k,J} - w^k \nabla log(\pi_{\lambda_k}(a|s)))$. Here the expectation is with respect to stationary state action distribution corresponding to $\pi_{\lambda_k}$. But we do not have samples of states action pairs from the stationary distribution with respect to the policy $\pi_{\lambda_k}$, we only have samples from the Markov chain induced by the policy $\pi_{\lambda_k}$. We thus refer to the theory in Doan (2022) and Assumption 3 to upper bound the error incurred.

For the error incurred in the actor update we define the related loss function as

**Definition 8.** *For iteration $k$ of the outer for loop of Algorithm 1 ,we define $w_k$ as the estimate of the minima of the loss function given by $\mathbb{E}_{(s,a)\sim\zeta_\nu^{\pi_{\lambda_k}}(s,a)}(A_{k,J}(s,a) - (w)\nabla log(\pi_{\lambda_k})(a|s))^2$ obtained at the end of the second inner for loop of Algorithm 1. We further define the true minima as*

$$w_k^* = \arg\min_w \mathbb{E}_{(s,a)\sim\zeta_\nu^{\pi_{\lambda_k}}(s,a)}(A_{k,J}(s,a) - (w)\nabla log(\pi_{\lambda_k})(a|s))^2, \tag{29}$$

For finding the estimate $w_k$, we re-use the state action pairs sampled in the first inner for loop of Algorithm 1. The difference between our estimate $w_k$ and the $w_k^*$ (which is also the minimizer of $II$) is then used to upper bound the difference between the value of $II$ at our estimate $w_k$ and the minimum possible value of $II$ achieved at $w_k^*$ which is upper bounded using Assumption 3. Details of this are given in lemma 7 in Appendix B.

## 7 CONCLUSIONS

In this paper, we study a natural actor critic algorithm with a neural network used to represent both the actor and the critic and find the sample complexity guarantees for the algorithm. We show that our approach achieves a sample complexity of $\tilde{\mathcal{O}}(\epsilon^{-4}(1-\gamma)^{-4})$. This demonstrates the first approach for achieving sample complexity beyond linear MDP assumptions for the critic.

