APPENDIX

## A    SUPPLEMENTARY LEMMAS

Here we provide some definitions and results that will be used to prove the lemmas stated in the paper.

**Definition 9.** *For a given set $Z \subset \mathbb{R}^n$, we define the Rademacher complexity of the set $Z$ as*

$$Rad(Z) = \mathbb{E}\left(\sup_{z \in Z} \frac{1}{n} \sum_{i=1}^{d} \Omega_i z_i\right) \tag{30}$$

*where $\Omega_i$ is random variable such that $P(\Omega_i = 1) = \frac{1}{2}$, $P(\Omega_i = -1) = \frac{1}{2}$ and $z_i$ are the co-ordinates of $z$ which is an element of the set $Z$*

**Lemma 1.** *Consider a set of observed data denoted by $z = \{z_1, z_2, \cdots z_n\} \in \mathbb{R}^n$, a parameter space $\Theta$, a loss function $\{l : \mathbb{R} \times \Theta \to \mathbb{R}\}$ where $0 \leq l(\theta, z) \leq 1 \ \forall (\theta, z) \in \Theta \times \mathbb{R}$. The empirical risk for a set of observed data as $R(\theta) = \frac{1}{n} \sum_{i=1}^{n} l(\theta, z_i)$ and the population risk as $r(\theta) = \mathbb{E}l(\theta, \tilde{z}_i)$, where $\tilde{z}_i$ is a co-ordinate of $\tilde{z}$ sampled from some distribution over $Z$.*

*We define a set of functions denoted by $\mathcal{L}$ as*

$$\mathcal{L} = \{z \in Z \to l(\theta, z) \in \mathbb{R} : \theta \in \Theta\} \tag{31}$$

*Given $z = \{z_1, z_2, z_3 \cdots, z_n\}$ we further define a set $\mathcal{L} \circ z$ as*

$$\mathcal{L} \circ z = \{(l(\theta, z_1), l(\theta, z_2), \cdots, l(\theta, z_n)) \in \mathbb{R}^n : \theta \in \Theta\} \tag{32}$$

*Then, we have*

$$\mathbb{E}\sup_{\theta \in \Theta} |\{r(\theta) - R(\theta)\}| \leq 2\mathbb{E}\left(Rad(\mathcal{L} \circ z)\right) \tag{33}$$

*If the data is of the form $z_i = (x_i, y_i), x \in X, y \in Y$ and the loss function is of the form $l(a_\theta(x), y)$, is $L$ lipschitz and $a_\theta : \Theta \times X \to \mathbb{R}$, then we have*

$$\mathbb{E}\sup_{\theta \in \Theta} |\{r(\theta) - R(\theta)\}| \leq 2L\mathbb{E}\left(Rad(\mathcal{A} \circ \{x_1, x_2, x_3, \cdots, x_n\})\right) \tag{34}$$

*where*
$$\mathcal{A} \circ \{x_1, x_2, \cdots, x_n\} = \{(a(\theta, x_1), a(\theta, x_2), \cdots, a(\theta, x_n)) \in \mathbb{R}^n : \theta \in \Theta\} \tag{35}$$

The detailed proof of the above statement is given in (Rebeschini, 2022)[2]. The upper bound for $\mathbb{E}\sup_{\theta \in \Theta}(\{r(\theta) - R(\theta)\})$ is proved in the aforementioned reference. However, without loss of generality the same proof holds for the upper bound for $\mathbb{E}\sup_{\theta \in \Theta}(\{R(\theta) - r(\theta)\})$. Hence the upper bound for $\mathbb{E}\sup_{\theta \in \Theta} |\{r(\theta) - R(\theta)\}|$ can be established.

**Lemma 2.** *Consider three random random variable $x \in \mathcal{X}$ and $y, y' \in \mathcal{Y}$. Let $\mathbb{E}_{x,y}, \mathbb{E}_x$ and $\mathbb{E}_{y|x}$, $\mathbb{E}_{y'|x}$ denote the expectation with respect to the joint distribution of $(x, y)$, the marginal distribution of $x$, the conditional distribution of $y$ given $x$ and the conditional distribution of $y'$ given $x$ respectively . Let $f_\theta(x)$ denote a bounded measurable function of $x$ parameterised by some parameter $\theta$ and $g(x, y)$ be bounded measurable function of both $x$ and $y$.*

*Then we have*

$$\arg\min_{f_\theta} \mathbb{E}_{x,y}\left(f_\theta(x) - g(x, y)\right)^2 = \arg\min_{f_\theta}\left(\mathbb{E}_x\left(f_\theta(x) - \mathbb{E}_{y'|x}(g(x, y')|x)\right)^2\right) \tag{36}$$

---

[2]Algorithmic Foundations of Learning [Lecture Notes].https://www.stats.ox.ac.uk/~rebeschi/teaching/AFoL/22/

*Proof.* Denote the left hand side of Equation equation 36 as $\mathbb{X}_\theta$, then add and subtract $\mathbb{E}_{y|x}(g(x,y)|x)$ to it to get

$$\mathbb{X}_\theta = \arg\min_{f_\theta} \left( \mathbb{E}_{x,y} \left( f_\theta(x) - \mathbb{E}_{y'|x}(g(x,y')|x) + \mathbb{E}_{y'|x}(g(x,y')|x) - g(x,y) \right)^2 \right) \quad (37)$$

$$= \arg\min_{f_\theta} \left( \mathbb{E}_{x,y} \left( f_\theta(x) - \mathbb{E}_{y'|x}(g(x,y')|x) \right)^2 + \mathbb{E}_{x,y} \left( y - \mathbb{E}_{y'|x}(g(x,y')|x) \right)^2 \right.$$
$$\left. - 2\mathbb{E}_{x,y}\Big( f_\theta(x) - \mathbb{E}_{y'|x}(g(x,y')|x) \Big) \Big( g(x,y) - \mathbb{E}_{y'|x}(g(x,y')|x) \Big) \right). \quad (38)$$

Consider the third term on the right hand side of Equation equation 38

$$2\mathbb{E}_{x,y} \left( f_\theta(x) - \mathbb{E}_{y'|x}(g(x,y')|x) \right) \left( g(x,y) - \mathbb{E}_{y'|x}(g(x,y')|x) \right)$$

$$= 2\mathbb{E}_x \mathbb{E}_{y|x} \left( f_\theta(x) - \mathbb{E}_{y'|x}(g(x,y')|x) \right) \left( g(x,y) - \mathbb{E}_{y'|x}(g(x,y')|x) \right) \quad (39)$$

$$= 2\mathbb{E}_x \left( f_\theta(x) - \mathbb{E}_{y'|x}(g(x,y')|x) \right) \mathbb{E}_{y|x} \left( g(x,y) - \mathbb{E}_{y'|x}(g(x,y')|x) \right) \quad (40)$$

$$= 2\mathbb{E}_x \left( f_\theta(x) - \mathbb{E}_{y'|x}(g(x,y')|x) \right) \left( \mathbb{E}_{y|x}(g(x,y)) - \mathbb{E}_{y|x} \left( \mathbb{E}_{y'|x}(g(x,y')|x) \right) \right) \quad (41)$$

$$= 2\mathbb{E}_x \left( f_\theta(x) - \mathbb{E}(y|x) \right) \left( \mathbb{E}_{y|x}(g(x,y)) - \mathbb{E}_{y'|x}(g(x,y')|x) \right) \quad (42)$$

$$= 0 \quad (43)$$

Equation equation 39 is obtained by writing $\mathbb{E}_{x,y} = \mathbb{E}_x \mathbb{E}_{y|x}$ from the law of total expectation. Equation equation 40 is obtained from equation 39 as the term $f_\theta(x) - \mathbb{E}_{y'|x}(g(x,y')|x)$ is not a function of $y$. Equation equation 41 is obtained from equation 40 as $\mathbb{E}_{y|x} \left( \mathbb{E}_{y'|x}(g(x,y')|x) \right) = \mathbb{E}_{y'|x}(g(x,y')|x)$ because $\mathbb{E}_{y'|x}(g(x,y')|x)$ is not a function of $y$ hence is constant with respect to the expectation operator $\mathbb{E}_{y|x}$.

Thus plugging in value of $2\mathbb{E}_{x,y} \left( f_\theta(x) - \mathbb{E}_{y'|x}(g(x,y')|x) \right) \left( g(x,y) - \mathbb{E}_{y'|x}(g(x,y')|x) \right)$ in Equation equation 38 we get

$$\arg\min_{f_\theta} \mathbb{E}_{x,y} \left( f_\theta(x) - g(x,y) \right)^2 = \arg\min_{f_\theta} (\mathbb{E}_{x,y} \left( f_\theta(x) - \mathbb{E}_{x,y'}(g(x,y')|x) \right)^2$$
$$+ \mathbb{E}_{x,y} \left( g(x,y) - \mathbb{E}_{y'|x}(g(x,y')|x) \right)^2). \quad (44)$$

Note that the second term on the right hand side of Equation equation 44 des not depend on $f_\theta(x)$ therefore we can write Equation equation 44 as

$$\arg\min_{f_\theta} \mathbb{E}_{x,y} \left( f_\theta(x) - g(x,y) \right)^2 = \arg\min_{f_\theta} \left( \mathbb{E}_{x,y} \left( f_\theta(x) - \mathbb{E}_{y'|x}(g(x,y')|x) \right)^2 \right) \quad (45)$$

Since the right hand side of Equation equation 45 is not a function of $y$ we can replace $\mathbb{E}_{x,y}$ with $\mathbb{E}_x$ to get

$$\arg\min_{f_\theta} \mathbb{E}_{x,y} \left( f_\theta(x) - g(x,y) \right)^2 = \arg\min_{f_\theta} \left( \mathbb{E}_x \left( f_\theta(x) - \mathbb{E}_{y'|x}(g(x,y')|x) \right)^2 \right) \quad (46)$$

$$\square$$

## B    SUPPORTING LEMMAS

We will now state the key lemmas that will be used for finding the sample complexity of the proposed algorithm.

**Lemma 3.** *For a given iteration $k$ of the outer for loop and iteration $j$ of the first inner for loop of Algorithm 1, the approximation error denoted by $\epsilon_{k,j}^1$ in Definition 4, we have*

$$\mathbb{E}\left(|\epsilon_{k,j}^1|\right) \leq \sqrt{\epsilon_{approx}}, \tag{47}$$

Where the expectation is with respect to and $(s,a) \sim \zeta_\nu^{\pi_{\lambda_k}}(s,a)$

*Proof Sketch:* We use Assumption 4 and the definition of the variance of a random variable to obtain the required result. The detailed proof is given in Appendix D.1.

**Lemma 4.** *For a given iteration $k$ of the outer for loop and iteration $j$ of the first inner for loop of Algorithm 1, $Q_{k,j}^1 = Q_{k,j}^2$, or equivalently $\epsilon_{k,j}^2 = 0$*

*Proof Sketch:* We use Lemma 2 in Appendix A and use the definitions of $Q_{k,j}^1$ and $Q_{k,j}^2$ to prove this result. The detailed proof is given in Appendix D.2.

**Lemma 5.** *For a given iteration $k$ of the outer for loop and iteration $j$ of the first inner for loop of Algorithm 1, if the number of state action pairs sampled are denoted by $n_{k,j}$, then the error $\epsilon_{k,j}^3$ defined in Definition 6 is upper bounded as*

$$\mathbb{E}\left(|\epsilon_{k,j}^3|\right) \leq \tilde{\mathcal{O}}\left(\frac{1}{\sqrt{n}}\right), \tag{48}$$

Where the expectation is with respect to and $(s,a) \sim \zeta_\nu^{\pi_{\lambda_k}}(s,a)$

*Proof Sketch:* First we note that For a given iteration $k$ of Algorithm 1 and iteration $j$ of the first for loop of Algorithm 1, $\mathbb{E}(R_{X_{k,j},Q_{k,j-1}}(\theta)) = L_{Q_{j,k-1}}(\theta)$ where $R_{X_{k,j},Q_{j,k-1}}(\theta)$ and $L_{Q_{j,k-1}}(\theta)$ are defined in Appendix D.3. We use this to get a probabilistic bound on the expected value of $|(Q_{j,k}^2) - (Q_{j,k}^3)|$ using Rademacher complexity theory when the samples are drawn from an ergodic Markov chain. The detailed proof is given in Appendix D.3. Note the presence of the $log(log(n_{k,j}))$ term is due to the fact that the state action samples belong to a Markov Chain.

**Lemma 6.** *For a given iteration $k$ of the outer for loop and iteration $j$ of the first inner for loop of Algorithm 1, let the number of steps of the gradient descent performed by Algorithm 1, denoted by $T_{k,j}$, the minimum number of neurons $m$ satisfy $m \geq \mathcal{O}(\delta^{-1})$ and the gradient descent step size $\alpha$ satisfy*

$$\alpha = \Theta\left(\frac{1}{poly(n,L).m}\right), \tag{49}$$

*Then with probability at least $1 - \delta$ the error $\epsilon_{k_4}$ defined in Definition 7 is upper bounded as*

$$\mathbb{E}(|\epsilon_{k,j}^4|) \leq \mathcal{O}\left(1 - \Omega\left(\frac{\alpha m}{n^2}\right)\right)^{T_{k,j}}, \tag{50}$$

Where the expectation is with respect to $(s,a) \sim \zeta_\nu^{\pi_{\lambda_k}}(s,a)$.

*Proof Sketch:* We use This is Theorem 5 from Zhu & Xu (2021) to prove this lemma. The detailed proof is given in Appendix D.4.

**Lemma 7.** *For a given iteration $k$ of the outer for loop of Algorithm 1, if the number of samples of the state action pairs sampled at each iteration of the first inner for loop are denoted by $n$ and $\beta_i$ be the step size in the gradient descent at iteration $i$ of the second inner for loop of Algorithm 1 which satisfies*

$$\beta_i = \frac{2}{\mu_k(i+1)}, \tag{51}$$

*where $\mu_k$ is the strong convexity parameter of the loss function $F_k = \mathbb{E}_{s,a \sim \zeta_\nu^{\pi_{\lambda_k}}(s,a)}(A_{k,J} - w^k \nabla log(\pi^{\lambda_k}(a|s)))^2$. Then, after $J.n$ iterations of gradient descent it holds that,*

$$(F_k(w_i)) \leq \tilde{\mathcal{O}}\left(\frac{log(J.n)}{J.n}\right) + F_k^*. \tag{52}$$

*where $F_k^* = \arg\min_w F_k(w)$.*

*Proof Sketch:* Note that we don not have access to state action samples belonging to the stationary state action distribution corresponding to the policy $\pi_{\lambda_k}$. We only have access to samples from Markov chain with the same stationary state action distribution. To account for this, we use the results in Doan (2022) and obtain the difference between the optimal loss function and the loss function obtained by performing stochastic gradient descent with samples from a Markov chain. The detailed proof is given in Appendix

## C  PROOF OF THEOREM 1

*Proof.* From Assumption 1, we have

$$
\log \frac{\pi_{\lambda_{k+1}}(a|s)}{\pi_{\lambda_k}(a|s)} \geq \nabla_{\lambda_k} \log \pi_{\lambda_k}(a|s) \cdot (\lambda^{k+1} - \lambda^k) - \frac{\beta}{2}||\lambda^{k+1} - \lambda^k||_2^2 \tag{53}
$$

$$
= \eta \log \pi_{\lambda_k}(a|s) \cdot w^k - \eta^2 \frac{\beta}{2}||w^k||_2^2 \tag{54}
$$

From the definition of KL divergence and from the performance difference lemma from (Kakade & Langford, 2002) we have

$$
\mathbb{E}_{s \sim d_\nu^{\pi^*}} \left( KL(\pi^*||\pi^{\lambda_k}) - \pi^*||\pi^{\lambda_{k+1}}) \right) = \mathbb{E}_{s \sim d_\nu^{\pi^*}} \mathbb{E}_{a \sim \pi^*(.|s)} \left[ log \frac{\pi^{\lambda_{k+1}}(a|s)}{\pi_{\lambda_k}(a|s)} \right] \tag{55}
$$

$$
\geq \eta \mathbb{E}_{s \sim d_\nu^{\pi^*}} \mathbb{E}_{a \sim \pi^*(.|s)} \left[ \nabla_{\lambda_k} \log \pi_{\lambda_k}(a|s) \cdot w^k \right] - \eta^2 \frac{\beta}{2}||w^k||_2^2 \tag{56}
$$

$$
= \eta \mathbb{E}_{s \sim d_\nu^{\pi^*}} \mathbb{E}_{a \sim \pi^*(.|s)} \left[ A^{\pi^{\lambda_k}} \right] - \eta^2 \frac{\beta}{2}||w^k||_2^2
$$
$$
+ \eta \mathbb{E}_{s \sim d_\nu^{\pi^*}} \mathbb{E}_{a \sim \pi^*(.|s)} \left[ \nabla_{\lambda_k} \log \pi_{\lambda_k}(a|s) \cdot w^k - A^{\pi^{\lambda_k}}(s,a) \right] \tag{57}
$$

$$
= (1 - \gamma) \eta \left( V^{\pi^*}(\nu) - V^k(\nu) \right) - \eta^2 \frac{\beta}{2}||w^k||_2^2 - \eta \cdot err_k. \tag{58}
$$

Equation equation 58 is obtained from Equation equation 58 using the performance difference lemma form (Kakade & Langford, 2002) where $A^{\pi^{\lambda_k}}$ is the advantage function to the corresponding to the policy $\pi^{\lambda_k}$.

Rearranging, we get

$$
\left( V^{\pi^*}(\nu) - V^k(\nu) \right) \leq \frac{1}{1 - \gamma} \left( \frac{1}{\eta} \mathbb{E}_{s \sim d_\nu^{\pi^*}} \left( KL(\pi^*||\pi^{\lambda_k}) - KL(\pi^*||\pi^{\lambda_{k+1}}) \right) + \eta^2 \frac{\beta}{2} \cdot W^2 + \eta \cdot err_k \right) \tag{59}
$$

Summing from 1 to $K$ and dividing by $K$ we get

$$\frac{1}{K}\sum_{k=1}^{K}\left(V^{\pi^*}(\nu) - V^k(\nu)\right) \leq \left(\frac{1}{1-\gamma}\right)\frac{1}{K}\sum_{k=1}^{K}\left(\mathbb{E}_{s\sim d_\nu^{\pi^*}}\left(KL(\pi^*||\pi^{\lambda_k}) - KL(\pi^*||\pi^{\lambda_{k+1}})\right) + \eta\cdot err_k\right)$$

$$+ \left(\frac{1}{1-\gamma}\right)\eta^2\frac{\beta}{2}\cdot W^2 \tag{60}$$

$$\leq \frac{1}{\eta(1-\gamma)}\frac{1}{K}\mathbb{E}_{s\sim\tilde{d}}\left(KL(\pi^*||\pi^{\lambda_0})\right) + \frac{\eta\beta\cdot W^2}{2(1-\gamma)} + \frac{1}{K(1-\gamma)}\sum_{k=1}^{K}err_k \tag{61}$$

$$\leq \frac{\log(|\mathcal{A}|)}{K\eta(1-\gamma)} + \frac{\eta^2\beta\cdot W^2}{2(1-\gamma)} + \frac{1}{K(1-\gamma)}\sum_{k=1}^{K}err_k \tag{62}$$

If we set $\eta = \frac{1}{\sqrt{K}}$ in Equation equation 62 we get

$$\frac{1}{K}\sum_{k=1}^{K}\left(V^{\pi^*}(\nu) - V^k(\nu)\right) \leq \frac{1}{\sqrt{K}}\left(\frac{2\log(|\mathcal{A}|) + \beta\cdot W^2}{2(1-\gamma)}\right) + \frac{1}{K(1-\gamma)}\sum_{k=1}^{K}err_k \tag{63}$$

Now we can redefine the term $err_k$ as $err_k = \mathbb{E}_{s,a}(|A^{\pi_{\lambda_k}} - w^k\nabla log(\pi^{\theta_k}(a|s))|)$ and the above inequality will still hold as $\mathbb{E}(x) \leq \mathbb{E}(|x|)$ for any random variable $x$. Now as in Equation equation 22 we have

$$err_k = \mathbb{E}_{s,a}(|A^{\pi_{\lambda_k}} - w^k\nabla log(\pi^{\theta_k}(a|s))|) \tag{64}$$

$$\leq \mathbb{E}_{s,a}(|A^{\pi_{\lambda_k}} - A_{k,J}|) + \mathbb{E}_{s,a}(|A_{k,J} - w^k\nabla log(\pi^{\theta_k}(a|s))|)$$

$$\leq \underbrace{\mathbb{E}_{s,a}(|A^{\pi_{\lambda_k}} - A_{k,J}|)}_{I} + \underbrace{\mathbb{E}_{s,a}(|A_{k,J} - w^k\nabla log(\pi^{\theta_k}(a|s))|)}_{II} \tag{65}$$

where $A_{k,j}$ is the estimate of $A^{\pi_{\lambda_k}}$ obtained at the $k^{th}$ iteration of Algorithm 1 and $s \sim d_\nu^{\pi^*}, a \sim \pi^*$.

We first derive bounds on $I$. From the definition of advantage function we have

$$\mathbb{E}(|A^{\pi_{\lambda_k}}(s,a) - A_{k,J}(s,a)|) = \mathbb{E}_{s\sim d_\nu^{\pi^*},a\sim\pi^*}|Q^{\pi_{\lambda_k}}(s,a) - E_{a_{i+1}\sim\pi^{\lambda_k}}Q^{\pi_{\lambda_k}}(s,a_{i+1})$$

$$-Q_{k,J}(s,a) + E_{a_{i+1}\sim\pi^{\lambda_k}}Q_{k,J}(s,a_{i+1})| \tag{66}$$

$$= \mathbb{E}_{s\sim d_\nu^{\pi^*},a\sim\pi^*}|Q^{\pi_{\lambda_k}}(s,a) - E_{a_{i+1}\sim\pi^{\lambda_k}}Q^{\pi_{\lambda_k}}(s,a_{i+1})$$

$$-Q_{k,J}(s,a) + E_{a_{i+1}\sim\pi^{\lambda_k}}Q_{k,J}(s,a_{i+1})| \tag{67}$$

$$\leq \mathbb{E}_{s\sim d_\nu^{\pi^*},a\sim\pi^*}|Q^{\pi_{\lambda_k}}(s,a) - Q_{k,J}(s,a)|$$

$$+ |\mathbb{E}_{s\sim d_\nu^{\pi^*},a_{i+1}\sim\pi^{\lambda_k}}|Q^{\pi_{\lambda_k}}(s,a) - Q_{k,J}(s,a)| \tag{68}$$

We write the second term on the right hand side of Equation equation 68 as $\int(|(Q^{\pi_{\lambda_k}}(s,a) - Q_{k,J}|(s,a))d(\mu_k)$ where $\mu_k$ is the measure associated with the state action distribution given by $s \sim d_\nu^{\pi^*}, a \sim \pi^{\lambda_k}$. Then we have

$$\int|Q^{\pi_{\lambda_k}}(s,a) - Q_{k,J}(s,a)|d(\mu_k) \leq \left\|\frac{d\mu_k}{d\mu^*}\right\|_\infty \int|Q^{\pi_{\lambda_k}}(s,a) - Q_{k,J}(s,a)|d(\mu^*) \tag{69}$$

where $\mu^*$ is the measure associated with the state action distribution given by $s \sim d_\nu^{\pi^*}, a_{i+1} \sim \pi^*$.

Now before we proceed further, we would like to introduce some notation for convenience, for two

probability measures $\mu_1$ and $\mu_2$ on we define $\left\|\frac{d\mu_1}{d\mu_2}\right\|_\infty = \phi_{\mu_1,\mu_2}$

Thus Equation equation 69 becomes

$$\int |Q^{\pi_{\lambda_k}}(s,a) - Q_{k,J}(s,a)| d(\mu_k) \le (\phi_{\mu_k, \mu^*}) \int |Q^{\pi_{\lambda_k}}(s,a) - Q_{k,J}(s,a)| d(\mu^*) \tag{70}$$

Since $\int |Q^{\pi_{\lambda_k}}(s,a) - Q_{k,J}(s,a)| d(\mu^*) = \mathbb{E}_{s \sim d_\nu^{\pi^*}, a_{i+1} \sim \pi^*} |Q^{\pi_{\lambda_k}}(s,a) - Q_{k,J}(s,a)|$

Equation equation 68 now becomes.

$$\mathbb{E}|A^{\pi_{\lambda_k}}(s,a) - A_{k,J}(s,a)| \le (1 + \phi_{\mu_k, \mu^*}) \mathbb{E}_{s \sim d_\nu^{\pi^*}, a \sim \pi^*} |Q^{\pi_{\lambda_k}}(s,a) - Q_{k,J}(s,a)| \tag{71}$$

Therefore minimizing $|A^{\pi_{\lambda_k}}(s,a) - A_{k,J}(s,a)|$ is equivalent to minimizing $|Q^{\pi_{\lambda_k}}(s,a) - Q_{k,J}(s,a)|$.

In order to prove the bound on $\mathbb{E}_{s \sim d_\nu^{\pi^*}, a \sim \pi^*} |Q^{\pi_{\lambda_k}}(s,a) - Q_{k,J}(s,a)|$ we first define some notation, let $Q_1, Q_2$ be two real valued functions on the state action space. The expression $Q_1 \ge Q_2$ implies $Q_1(s,a) \ge Q_2(s,a) \ \forall (s,a) \in \mathcal{S} \times \mathcal{A}$.

Let $Q_{k,j}$ denotes our estimate of the action value function at iteration $k$ of Algorithm 1 and iteration $j$ of the first for loop of Algorithm 1. $Q^{\pi_{\lambda_k}}$ denotes the action value function induced by the policy $\pi_{\lambda_k}$.

Consider $\epsilon_{k,j+1} = T^{\pi_{\lambda_k}} Q_{k,j} - Q_{k,j+1}$.

Thus we get,

$$Q^{\pi_{\lambda_k}} - Q_{k,j+1} = T^{\pi_{\lambda_k}} Q^{\pi_{\lambda_k}} - Q_{k,j+1} \tag{72}$$
$$= T^{\pi_{\lambda_k}} Q^{\pi_{\lambda_k}} - T^{\pi_{\lambda_k}} Q_{k,j} + T^{\pi_{\lambda_k}} Q_{k,j}$$
$$- Q_{k,j+1} \tag{73}$$
$$= \gamma(P^{\pi_{\lambda_k}}(Q^{\pi_{\lambda_k}} - Q_{k,j})) + \epsilon_{k,j+1} \tag{74}$$
$$|Q^{\pi_{\lambda_k}} - Q_{k,j+1}| \le \gamma(P^{\pi_{\lambda_k}}(|Q^{\pi_{\lambda_k}} - Q_{k,j}|)) + |\epsilon_{k,j+1}| \tag{75}$$

Right hand side of Equation equation 72 is obtained by writing $Q^{\pi_{\lambda_k}} = T^{\pi_{\lambda_k}} Q^{\pi_{\lambda_k}}$. This is because the function $Q^{\pi_{\lambda_k}}$ is a stationary point with respect to the operator $T^{\pi_{\lambda_k}}$. Equation equation 73 is obtained from equation 72 by adding and subtracting $T^{\pi^{\lambda_k}}$. We get equation 75 from equation 74 by taking the absolute value on both sides and applying the triangle inequality on the right hand side.

By recursion on $k$, we get,

$$|Q^{\pi_{\lambda_k}} - Q_{k,J}| \le \sum_{j=0}^{J-1} \gamma^{J-j-1}(P^{\pi_{\lambda_k}})^{J-j-1} |\epsilon_{k,j+1}| + \gamma^J (P^{\pi_{\lambda_k}})^J (|Q^{\pi_{\lambda_k}} - Q_0|) \tag{76}$$

From this we obtain

$$\mathbb{E}_{s \sim d_\nu^{\pi^*}, a \sim \pi^*} |Q^{\pi_{\lambda_k}} - Q_{k,J}| \le \sum_{k=0}^{J-1} \gamma^{J-j-1} \mathbb{E}_{s \sim d_\nu^{\pi^*}, a \sim \pi^*} ((P^{\pi_{\lambda_k}})^{K-J-1} |\epsilon_{k,j+1}|)$$
$$+ \gamma^J \mathbb{E}_{s \sim d_\nu^{\pi^*}, a \sim \pi^*} (P^{\pi_{\lambda_k}})^J (|Q^{\pi_{\lambda_k}} - Q_0|) \tag{77}$$

For a fixed $j$ consider the term $\mathbb{E}_{s \sim d_\nu^{\pi^*}, a \sim \pi^*} ((P^{\pi_{\lambda_k}})^{J-j-1} |\epsilon_{k,j+1}|)$. We then write

$$\mathbb{E}_{s \sim d_\nu^{\pi^*}, a \sim \pi_j} ((P^{\pi_{\lambda_k}})^{J-j-1} |\epsilon_{k,j+1}|) \le \left\| \frac{d(P^{\pi_{\lambda_k}})^{J-j-1} \mu_j}{d\mu_k'} \right\|_\infty \int |\epsilon_{k,j+1}| \, d\mu_k' \tag{78}$$

$$\le (\phi_{\mu_k', \mu_j}) \mathbb{E}_{(s,a) \sim \zeta_\nu^{\pi_{\lambda_k}}(s,a)} (|\epsilon_{k,j+1}|) \tag{79}$$

Here $\mu_j$ is the measure associated with the state action distribution given by sampling from $s \sim d_\nu^{\pi^*}$, $a_{i+1} \sim \pi^*$ and then applying the operator $P^{\pi_{\lambda_k}}$, $J - j - 1$ times. $\mu_j$ is the measure associated with the steady state action distribution given by $(s, a) \sim \zeta_\nu^{\pi_{\lambda_k}}(s, a)$. Thus Equation equation 77 becomes

$$\mathbb{E}_{s \sim d_\nu^{\pi^*}, a \sim \pi^*} |Q^{\pi_{\lambda_k}} - Q_{k,J}| \leq \sum_{k=0}^{J-1} \gamma^{J-j-1} (\phi_{\mu_k', \mu_j}) \mathbb{E}_{(s,a) \sim \zeta_\nu^{\pi_{\lambda_k}}(s,a)} (|\epsilon_{k,j+1}|)| + \gamma^J \left( \frac{R_{max}}{1 - \gamma} \right) \tag{80}$$

We get the second term on the right hand side by noting that $(Q^{\pi_{\lambda_k}} - Q_0) \leq \frac{R_{max}}{1-\gamma}$. Now splitting $\epsilon_{k,j+1}$ as was done in Equation equation 25 we obtain

$$\mathbb{E}_{s \sim d_\nu^{\pi^*}, a \sim \pi^*} |Q^{\pi_{\lambda_k}} - Q_{k,J}| \leq \sum_{j=0}^{J-1} \gamma^{J-j-1} \big( (\phi_{\mu_k', \mu_j}) \mathbb{E}|\epsilon_{k,j+1}^1| + (\phi_{\mu_k', \mu_j}) \mathbb{E}|\epsilon_{k,j+1}^2|$$
$$+ (\phi_{\mu_k', \mu_j}) \mathbb{E}|\epsilon_{k,j+1}^3| + (\phi_{\mu_k', \mu_j}) \mathbb{E}|\epsilon_{k,j+1}^4| \big) + \gamma^J \left( \frac{R_{max}}{1 - \gamma} \right) \tag{81}$$

Now using Lemmas 3, 4, 5, 6 we have for $m \geq \mathcal{O}(K.J.\delta^{-1})$ with probability at-least $1 - \delta$ we have

$$\mathbb{E}_{s \sim d_\nu^{\pi^*}, a \sim \pi^*} (Q^{\pi_{\lambda_k}} - Q_{k,j}) \leq \sum_{j=0}^{J-1} \left( \mathcal{O}\left( \frac{1}{\sqrt{n}} \right) + \mathcal{O}(\sqrt{\epsilon_{approx}}) + \mathcal{O}\left( 1 - \Omega\left( \frac{\alpha m}{n^2} \right) \right)^n \right.$$
$$\left. + \mathcal{O}(\gamma^J) \right) \tag{82}$$

Note we had to increase the requirement of $m$ from $m \geq \mathcal{O}(\delta^{-1})$ to $m \geq \mathcal{O}(K.J.\delta^{-1})$ as we want the probability statement of lemma 6 to hold for all iterations of the outer for loop and the first inner for loop. Also note that for lemma 5 the number of iterations of gradient descent is $n$, so we replace $T_{k,j}$ to $n$.

From Equation equation 71 we get that for we have for $m \geq \mathcal{O}(K.J.\delta^{-1})$ with probability at-least $1 - \delta$ we have

$$\mathbb{E}_{s \sim d_\nu^{\pi^*}, a \sim \pi^*} (A^{\pi_{\lambda_k}} - A_{k,j}) \leq \sum_{j=0}^{J-1} \left( \mathcal{O}\left( \frac{1}{\sqrt{n}} \right) + \mathcal{O}(\sqrt{\epsilon_{approx}}) + \mathcal{O}\left( 1 - \Omega\left( \frac{\alpha m}{n^2} \right) \right)^n \right.$$
$$\left. + \mathcal{O}(\gamma^J) \right) \tag{83}$$

We now derive bounds on $II$. Note that $II$ can be upper bounded as

$$\mathbb{E}_{s \sim d_\nu^{\pi^*}, a \sim \pi^*} (|A_{k,J} - w^k \nabla log(\pi^{\theta_k}(a|s))|) \leq (\phi_{\alpha_k, \mu^*}) \mathbb{E}_{s,a \sim \zeta_\nu^{\pi_{\lambda_k}}} (|A_{k,J} - w^k \nabla log(\pi^{\theta_k}(a|s))|) \tag{84}$$

Here $\alpha_k$ is the measure corresponding to $s, a \sim \zeta_\nu^{\pi_{\lambda_k}}(s, a)$ and $\mu^*$ is as defined previously.

Now from lemma if we have $\beta_{i,k} = \frac{2}{\mu_k(i+1)}$, where $\mu_k$ is the strong convexity parameter for the loss function in $F_k(w) = \mathbb{E}_{s,a \sim \zeta_\pi^\nu} (|A_{k,J} - w^k \nabla log(\pi^{\theta_k}(a|s))|)^2$, we obtain from lemma 7 that

$$||w_k - w_k^*||_2 \quad \leq \quad \mathcal{O}\left(\frac{\log(n)}{n}\right) \tag{85}$$

Now define the function . From this definition and the fact that $w_k^*$ is also the minimizer of $II$ we obtain

$$F_k(w_k) - F_k(w^*) \leq l_{F_k}||w_k - w^*||_2 \quad \leq \quad \mathcal{O}\left(\frac{\log(n)}{n}\right) \tag{86}$$

where $l_{F_k}$ is lipschitz constant of $F_k(w)$. Thus we obtain

$$F_k(w_k) - F_k(w^*) \quad \leq \quad \mathcal{O}\left(\frac{\log(n)}{n}\right) \tag{87}$$

which gives us

$$F_k(w_k) \quad \leq \quad \mathcal{O}\left(\frac{\log(n)}{n}\right) + \epsilon_{bias} \tag{88}$$

We get equation 88 from equation 87 by using assumption 4.

Now from equation 84 we get

$$II \quad \leq \quad \mathcal{O}\left(\frac{\log(n)}{n}\right) + \mathcal{O}(\epsilon_{bias}) \tag{89}$$

Plugging Equations equation 88 and equation 83 in Equation equation 63 we get for $m \geq \mathcal{O}(K.J.\delta^{-1})$ with probability at-least $1 - \delta$ we have

$$\min_{k \leq K}(V^*(\nu) - V^{\pi_{\lambda_K}}(\nu)) \leq \frac{1}{K}\sum_{k=1}^{K}(V^*(\nu) - V^{\pi_{\lambda_K}}(\nu)) \tag{90}$$

$$\leq \mathcal{O}\left(\frac{1}{\sqrt{K}(1-\gamma)}\right) + \frac{1}{K(1-\gamma)}\sum_{k=1}^{K}\left(\mathcal{O}\left(\frac{\log(J.n)}{J.n}\right) + \mathcal{O}(\gamma^J)\right)$$

$$+ \frac{1}{K(1-\gamma)}\sum_{k=1}^{K}\sum_{j=0}^{J-1}\left(\mathcal{O}\left(1 - \Omega\left(\frac{\alpha m}{n^2}\right)\right)^n + \mathcal{O}\left(\frac{1}{\sqrt{n}}\right)\right)$$

$$+ \frac{1}{1-\gamma}\left(\mathcal{O}(\epsilon_{bias}) + \mathcal{O}(\sqrt{\epsilon_{approx}})\right). \tag{91}$$

$\square$

## D   PROOF OF SUPPORTING LEMMAS

### D.1   PROOF OF LEMMA 3

*Proof.* Using Assumption 3 and the definition of $Q_{kj_1}$ for some iteration $k$ of Algorithm 1 we have

$$\mathbb{E}_{s,a}(T^{\pi_{\lambda_k}}Q_{k,j-1} - Q_{k,j}^1)^2 \leq \epsilon_{approx} \tag{92}$$

where $(s, a) \sim \zeta_\nu^{\pi_{\lambda_k}}(s, a)$.

Since $|a|^2 = a^2$ we obtain

$$\mathbb{E}(|T^{\pi_{\lambda_k}} Q_{k,j-1} - Q_{k,j}^1|)^2 \leq \epsilon_{approx} \tag{93}$$

We have for a random variable $x$, $Var(x) = \mathbb{E}(x^2) - (\mathbb{E}(x))^2$ hence $\mathbb{E}(x) = \sqrt{\mathbb{E}(x^2) - Var(x)}$, Therefore replacing $x$ with $|T^{\pi_{\lambda_k}} Q^{\pi_{\lambda_k}} - Q_{k1}|$ we get

using the definition of the variance of a random variable we get

$$\mathbb{E}(|T^{\pi_{\lambda_k}} Q_{k,j-1} - Q_{k,j}^1|) = \sqrt{\mathbb{E}(|T^{\pi_{\lambda_k}} Q_{k,j-1} - Q_{k,j}^1|)^2 - Var(|T^{\pi_{\lambda_k}} Q_{k,j-1} - Q_{k,j}^1|)} \tag{94}$$

$$\leq \sqrt{\mathbb{E}(|T^{\pi_{\lambda_k}} Q_{k,j-1} - Q_{k,j}^1|)^2} \tag{95}$$

Therefore by definition of $Q_{k,j}^1$ and assumption 4 we have

$$\mathbb{E}(T^{\pi_{\lambda_k}} Q_{k,j-1} - Q_{k,j}^1|) \leq \sqrt{\epsilon_{approx}} \tag{96}$$

Since $\epsilon_{k_1} = T^{\pi_{\lambda_k}} Q^{\pi_{\lambda_k}} - Q_{k1}$ we have

$$\mathbb{E}(|\epsilon_{k,j_1}|) \leq \sqrt{\epsilon_{approx}} \tag{97}$$

$\square$

## D.2 PROOF OF LEMMA 4

*Proof.* From Lemma 2, we have

$$\underset{f_\theta}{\arg\min} \, \mathbb{E}_{x,y} \left( f_\theta(x) - g(x,y) \right)^2 = \underset{f_\theta}{\arg\min} \left( \mathbb{E}_{x,y} \left( f_\theta(x) - \mathbb{E}(g(y^{'},x)|x) \right)^2 \right) \tag{98}$$

We label $x$ to be the state action pair $(s,a)$, $y$ is the next state action pair denoted by $(s^{'}, a_{i+1})$. The function $f_\theta(x)$ to be $Q_\theta(s,a)$ and $g(x,y)$ to be the function $r^{'}(s,a) + \gamma Q_{k,j-1}(s^{'}, a_{i+1})$

Then the loss function corresponding to Equation equation 36 becomes

$$\mathbb{E}(Q_\theta(s,a) - (r(s,a) + \gamma \mathbb{E} Q_{k,j-1}(s^{'}, a_{i+1})))^2 \tag{99}$$

where $(s,a) \sim \zeta_\nu^{\pi_{\lambda_k}}(s,a)$, $s^{'} \sim P(.|(s,a))$, $a_{i+1} \sim \pi^{\lambda_k}(.|s^{'})$ and $r(s,a) \sim R(.|s,a)$.

Therefore by Lemma 2, we have that the function $Q_\theta(s,a)$ which minimizes Equation equation 99 it will be minimizing

$$\mathbb{E}_{s \sim d_\nu^{\pi_{\lambda_k}}, a \sim \pi_{\lambda_k}} (Q_\theta(s,a) - \mathbb{E}_{s^{'} \sim P(s^{'}|s,a), r \sim \mathcal{R}(.|s,a)}(r(s,a) + \gamma \mathbb{E} Q_{k,j-1}(s^{'}, a_{i+1})|s,a))^2 \tag{100}$$

But we have from Equation that

$$\mathbb{E}_{s^{'} \sim P(s^{'}|s,a), r \sim R(.|s,a))}(r(s,a) + \gamma \mathbb{E} Q_{k,j-1}(s^{'}, a_{i+1})|s,a) = T^{\pi_{\lambda_k}} Q_{k,j-1} \tag{101}$$

Combining Equation equation 99 and equation 101 we get

$$\underset{Q_\theta}{\arg\min} \, \mathbb{E}(Q_\theta(s,a) - (r(s,a) + \gamma Q_{k,j-1}(s^{'}, a_{i+1})))^2 = \underset{Q_\theta}{\arg\min} \, \mathbb{E}(Q_\theta(s,a) - T^{\pi_{\lambda_k}} Q_{k,j-1})^2 \tag{102}$$

The left hand side of Equation equation 102 is $Q_{k,j}^2$ as defined in Definition 2 and the right hand side is $Q_{k,j}^1$ as defined in Definition 1, which gives us

$$Q_{k,j}^2 = Q_{k,j}^1 \tag{103}$$

□

### D.3    PROOF OF LEMMA 5

*Proof.* We define $R_{X_{k,j},Q_{k,j-1}}(\theta)$ as

$$R_{X_{k,j},Q_{k,j-1}}(\theta) = \frac{1}{n} \sum_{(s_i,a_i) \in X_{k,j}} \left( Q_\theta(s_i,a_i) - \left( r^{'}(s_i,a_i) + \gamma Q_{k,j-1}(s_{i+1},a_{i+1}) \right) \right)^2,$$

Here, where $s, a$ are sampled from a Markov chain whose stationary distribution is, $(s,a) \sim \zeta_\nu^{\pi_{\lambda_k}}(s,a)$. $Q_\theta$ is is the neural network corresponding to the parameter $\theta$ and $Q_{k,j-1}$ is the estimate of the $Q$ function obtained at iteration $k$ of the outer for loop and iteration $j-1$ of the first inner for loop of Algorithm 1.

We also define the term

$$L_{Q_{k,j-1}}(Q_\theta) = \mathbb{E}(Q_\theta(s,a) - (r'(s,a) + \gamma Q_{k,j-1}(s',a'))^2 \tag{104}$$

where $(s,a) \sim \zeta_{\pi_{\lambda_k}}^\nu, r'(\cdot|s,a) \sim R(\cdot|s,a), a_{i+1} \sim \pi_{\lambda_k}$

We denote by $\theta_{k,j}^2, \theta_{k,j}^3$ the parameters of the neural networks $Q_{k,j}^2, Q_{k,j}^3$ respectively. $Q_{k,j}^2, Q_{k,j}^3$ are defined in Definition 2 and 3 respectively.

We then obtain,

$$
\begin{aligned}
R_{X_{k,j},Q_{k,j-1}}(\theta_{k,j}^2) - R_{X_{k,j},Q_{k,j-1}}(\theta_{k,j}^3) &\leq R_{X_{k,j},Q_{k,j-1}}(\theta_{k,j}^2) - R_{X_{k,j},Q_{k,j-1}}(\theta_{k,j}^3) \\
&\quad + L_{Q_{k,j-1}}(\theta_{k,j}^2) - L_{Q_{k,j-1}}(\theta_{k,j}^3) \\
& \tag{105}\\
&= R_{X_{k,j},Q_{k,j-1}}(\theta_{k,j}^2) - L_{Q_{k,j-1}}(\theta_{k,j}^2) \\
&\quad - R_{X_{k,j},Q_{k,j-1}}(\theta_{k,j}^3) + L_{Q_{k,j-1}}(\theta_{k,j}^2) \\
& \tag{106}\\
&\leq \underbrace{|R_{X_{k,j},Q_{k,j-1}}(\theta_{k,j}^2) - L_{Q_{k,j-1}}(\theta_{k,j}^2)|}_{I} \\
&\quad + \underbrace{|R_{X_{k,j},Q_{k,j-1}}(\theta_{k,j}^3) - L_{Q_{k,j-1}}(\theta_{k,j}^3)|}_{II}
\end{aligned}
$$

$$\tag{107}$$

We get the inequality in Equation equation 105 because $L_{Q_{k,j-1}}(\theta_{k,j}^3) - L_{Q_{k,j-1}}(\theta_{k,j}^2) > 0$ as $Q_{k,j}^2$ is the minimizer of the loss function $L_{Q_{k,j-1}}(Q_\theta)$.

Consider Lemma 1. The loss function $R_{X_{k,j},Q_{k,j-1}}(\theta_{k,j}^3)$ can be written as the mean of loss functions of the form $l(a_\theta(s_i,a_i,s_{i+1},a_{i+1}), y_i)$ where $l$ is the square function. $a_\theta(s_i,a_i,s_{i+1},a_{i+1}) = Q_\theta(s_i,a_i)$ and $y_i = \left( r^{'}(s_i,a_i) + \gamma Q_{k,j-1}(s_{i+1},a_{i+1}) \right)$. Thus we have

$$\mathbb{E}\sup_{\theta \in \Theta'} |R_{X_{k,j},Q_{k,j-1}}(\theta) - L_{Q_{k,j-1}}(\theta)| \leq \tag{108}$$
$$2\eta^{'}\mathbb{E}\left(Rad(\mathcal{A} \circ \{(s_1,a_1,s_2,a_2),(s_2,a_2,s_3,a_3),\cdots,(s_{n-1},a_{n-1},s_n,a_n)\})\right)$$

Note that the expectation is over all $(s_i, a_i)$ and that the set $\Theta^{'} = \{Q_{\theta^2_{k,j}}, Q_{\theta^3_{k,j}}\}$. We use this set because we only need this inequality to hold for $Q_{\theta^2_{k,j}}$ and $Q_{\theta^3_{k,j}}$. Where $n = |X_{k,j}|$, $(\mathcal{A} \circ \{(s_1, a_1), (s_2, a_2), (s_3, a_3), \cdots, (s_n, a_n)\} = \{Q_\theta(s_1, a_1), Q_\theta(s_2, a_2), \cdots, Q_\theta(s_n, a_n)\}$ and $\eta_{i+1}$ is the Lipschitz constant for the square function over the state action space $[0, 1]^d$. The expectation is with respect to $(s, a) \sim \zeta^\nu_{\pi_{\lambda_k}}$, $s^{'}_i \sim P(s^{'}|s, a)$ $r_i \sim R(.|s_i, a_i)_{i \in (1, \cdots, n)}$.

Now from theorem 5 and theorem 1 of Bertail & Portier (2019) we have that

$$(Rad(\mathcal{A} \circ \{(s_1, a_1, s_2, a_2), (s_2, a_2, s_3, a_3), \cdots, (s_{n-1}, a_{n-1}, s_n, a_n)\})) \leq C_k \frac{1}{\sqrt{n}} \tag{109}$$

Note that in Bertail & Portier (2019) a factor of $\log \log(n)$ in the numerator is introduced in later theorems, we ignore that factor due to the fact that it is practically constant and the we will have a factor of $\log(n)$ from the error incurred in the actor step.

We use this result as the state action pairs are drawn not from the stationary state of the policy $\pi_{\lambda_k}$ but from a Markov chain with the same steady state distribution. Thus we have

$$\mathbb{E}|(R_{X_{k,j}, Q_{k,j-1}}(\theta^2_{k,j})) - L_{Q_{k,j-1}}(\theta^2_{k,j})| \leq C_k \frac{1}{\sqrt{n}} \tag{110}$$

The same argument can be applied for $Q^3_{k,j}$ to get

$$\mathbb{E}|(R_{X_{k,j}, Q_{k,j-1}}(\theta^3_{k,j})) - L_{Q_{k,j-1}}(\theta^3_{k,j})| \leq C_k \frac{1}{\sqrt{n}} \tag{111}$$

Then we have

$$\mathbb{E}\left(R_{X_{k,j}, Q_{k,j-1}}(\theta^2_{k,j}) - R_{X_{k,j}, Q_{k,j-1}}(\theta^3_{k,j})\right) \leq C_k \frac{1}{\sqrt{n}} \tag{112}$$

Plugging in the definition of $R_{X_{k,j}, Q_{k,j-1}}(\theta^2_{k,j}), R_{X_{k,j}, Q_{k,j-1}}(\theta^3_{k,j})$ in equation equation 112 and denoting $C_k \frac{1}{\sqrt{n}}$ as $\epsilon$ we get

$$\frac{1}{n} \sum_{i=1}^n \left( \mathbb{E}(Q^2_{k,j}(s_i, a_i) - (r^{'}(s_i, a_i) + \gamma Q^2_{k,j}(s_{i+1}, a_{i+1})))^2 \right.$$
$$\left. -\mathbb{E}(Q^3_{k,j}(s_i, a_i) - (r^{'}(s_i, a_i) + \gamma Q^3_{k,j}(s_{i+1}, a_{i+1})))^2 \right) \leq \epsilon \tag{113}$$

Now for a fixed $i$ consider the term $\alpha_i$ defined as.

$$\mathbb{E}_{s_{i+1} \sim P(.|s_i, a_i)}(Q^2_{k,j}(s_i, a_i) - (r^{'}(s_i, a_i) + \gamma Q^2_{k,j}(s_{i+1}, a_{i+1})))^2$$
$$-\mathbb{E}_{s_{i+1} \sim P(.|s_i, a_i)}(Q^3_{k,j}(s_i, a_i) - (r^{'}(s_i, a_i) + \gamma Q^3_{k,j}(s_{i+1}, a_{i+1})))^2 \tag{114}$$

where $s_i, a_i, s_{i+1}, a_{i+1}$ are drawn from the state action distribution at the $i^{th}$ step of the Markov chain induced by following the policy $\pi_{\lambda_k}$.

Now for a fixed $i$ consider the term $\beta_i$ defined as.

$$\mathbb{E}_{s_{i+1} \sim P(.|s_i, a_i)}(Q^2_{k,j}(s_i, a_i) - (r^{'}(s_i, a_i) + \gamma Q^2_{k,j}(s_{i+1}, a_{i+1})))^2$$
$$-\mathbb{E}_{s_{i+1} \sim P(.|s_i, a_i)}(Q^3_{k,j}(s_i, a_i) - (r^{'}(s_i, a_i) + \gamma Q^3_{k,j}(s_{i+1}, a_{i+1})))^2 \tag{115}$$

where $s_i, a_i, s_{i+1}, a_{i+1}$ are drawn from the steady state action distribution with $(s, a) \sim \zeta^\nu_{\pi_{\lambda_k}}$. Note here that $\alpha_i$ and $\beta_i$ are the same function with only the state action pairs being drawn from different distributions.

Using these definitions we obtain

$$
\begin{aligned}
|\mathbb{E}(\alpha_i) - \mathbb{E}(\beta_i)| \quad &\leq \quad \sup_{(s_i,a_i)} |2.\max(\alpha_i,\beta_i)|(\kappa_i) \quad\quad (116) \\
&\leq \quad \left(4\frac{R}{1-\gamma}\right)^2 p\rho^i \quad\quad\quad (117)
\end{aligned}
$$

We obtain Equation equation 116 by using the identity $|\int f d\mu - \int f d\nu| \leq |\max_{\mathcal{S}\times\mathcal{A}}(f)|\sup_{\mathcal{S}\times\mathcal{A}}\int(d\mu - d\nu)| \leq |\max_{\mathcal{S}\times\mathcal{A}}(f)|d_{TV}(\mu,\nu)|$, where $\mu$ and $\nu$ are two $\sigma$ finite state action probability measures and $f$ is a bounded measurable function. We have used $\kappa_i$ to represent the total variation distance between the state action measures of the steady state action distribution denoted by $(s,a) \sim \zeta^\nu_{\pi_{\lambda_k}}$ and the state action distribution at the $i^{th}$ step of the Markov chain induced by following the policy $\pi^{\lambda_k}$. The expectation is with respect to $(s_i,a_i)$. We obtain Equation equation 117 from Equation equation 116 by using Assumption 2 and the fact that $\alpha_i$ and $\beta_i$ are upper bounded by $\left(4\frac{R}{1-\gamma}\right)^2$

From equation equation 117 we get

$$
\mathbb{E}(\alpha_i) \quad \geq \quad \mathbb{E}(\beta_i) - 4\left(\frac{R}{1-\gamma}\right)^2 p\rho^i \quad\quad (118)
$$

We get Equation equation 118 from Equation equation 117 using the fact that $|a-b| \leq c$ implies that $(-c \geq (a-b) \leq c)$ which in turn implies $a \geq b - c$.

Using Equation equation 118 in equation equation 115 we get

$$
\begin{aligned}
&\frac{1}{n}\sum_{i=1}^{n}\left(\mathbb{E}(Q^2_{k,j}(s_i,a_i) - (r^{'}(s_i,a_i)+\gamma Q^2_{k,j}(s_{i+1},a_{i+1})))^2\right. \\
&\left.\quad -\mathbb{E}(Q^3_{k,j}(s_i,a_i) - (r^{'}(s_i,a_i)+\gamma Q^3_{k,j}(s_{i+1},a_{i+1})))^2\right) \\
&\leq \quad \epsilon + \frac{1}{n}\sum_{i=1}^{n}4\left(\frac{R}{1-\gamma}\right)^2 p\rho^i \\
&\leq \quad \epsilon + \frac{1}{n}4\left(\frac{R}{1-\gamma}\right)^2 p\frac{1}{1-\rho}
\end{aligned}
$$
$$(119)$$

In Equation equation 119 $(s_i,a_i)$ are now drawn from $(s,a) \sim \zeta^\nu_{\pi_{\lambda_k}}$ for all $i$.

We ignore the second term on the right hand side as it is $\tilde{\mathcal{O}}\left(\frac{1}{n}\right)$ as compared to the first term which is $\tilde{\mathcal{O}}\left(\frac{1}{\sqrt{n}}\right)$. Additionally the expectation in Equation equation 119 is with respect to $(s,a) \sim \zeta^\nu_{\pi_{\lambda_k}}, r'(\cdot|s,a) \sim R(\cdot|s,a), s_{i+1} \sim P(.|s_i,a_i), a_{i+1} \sim \pi^{\lambda_k}$

Since now we have $(s,a) \sim \zeta^\nu_{\pi_{\lambda_k}}$ for all $i$, Equation equation 119 is equivalent to,

$$
\mathbb{E}\underbrace{(Q^2_{k,j}(s,a) - Q^3_{k,j}(s,a))}_{A1}\underbrace{(Q^2_{k,j}(s,a) + Q^3_{k,j}(s,a) - 2(r^{'}(s,a)) + \gamma Q_{k,j-1}(s^{'},a_{i+1}))}_{A2} \quad \leq \quad \epsilon
$$
$$(120)$$

Where the expectation is now over $(s,a) \sim \zeta^{\nu}_{\pi_{\lambda_k}}$, $r^{'}(s,a) \sim R(.|s,a)$ and $s^{'} \sim P(.|s,a), a_{i+1} \sim \pi_{\lambda_k}$. We re-write Equation equation 120 as

$$\int \underbrace{(Q^2_{k,j}(s,a) - Q^3_{k,j}(s,a))}_{A1} \times$$

$$\times \underbrace{(Q^2_{k,j}(s,a) + Q^3_{k,j}(s,a) - 2(r^{'}(s,a)) + \gamma \max_{a \in \mathcal{A}} Q_{k,j-1}(s^{'},a))}_{A2} \times$$

$$\times d\mu_1(s,a)d\mu_2(r)d\mu_3(s^{'})d\mu_4(a_{i+1}) \leq \epsilon. \tag{121}$$

Where $\mu_1$ is the state action distribution $(s,a) \sim \zeta^{\nu}_{\pi_{\lambda_k}}$, $\mu_2$, $\mu_3$, $\mu_4$ are the measures with respect to $(s,a)$, $r^{'}$, $s^{'}$ and $a_{i+1}$ respectively.

Now for the integral in Equation equation 121 we split the integral into four different integrals. Each integral is over the set of $(s,a), r^{'}, s^{'}, a_{i+1}$ corresponding to the 4 different combinations of signs of $A1, A2$.

$$\int_{\{(s,a),r',s'\}:A1 \geq 0, A2 \geq 0} (A1)(A2)d\mu_1(s,a)d\mu_2(r)d\mu_3(s^{'})d\mu_4(a_{i+1})$$

$$+ \int_{\{(s,a),r',s'\}:A1 < 0, A2 < 0} (A1)(A2)d\mu_1(s,a)d\mu_2(r)d\mu_3(s^{'})d\mu_4(a_{i+1})$$

$$+ \int_{\{(s,a),r',s'\}:A1 \geq 0, A2 < 0} (A1)(A2)d\mu_1(s,a)d\mu_2(r)d\nu_3(s^{'})d\mu_4(a_{i+1})$$

$$+ \int_{\{(s,a),r',s'\}:A1 < 0, A2 \geq 0} (A1)(A2)d\mu_1(s,a)d\mu_2(r)d\mu_3(s^{'})d\mu_4(a_{i+1}) \leq \epsilon \tag{122}$$

Now note that the first 2 terms are non-negative and the last two terms are non-positive. We then write the first two terms as

$$\int_{\{(s,a),r',s'\}:A1 \geq 0, A2 \geq 0} (A1)(A2)d(s,a)d\mu_1(s,a)d\mu_2(r)d\mu_3(s^{'})d\mu_4(a_{i+1})$$

$$= C_{k,j_1} \int |Q^2_{k,j} - Q^3_{k,j}|d\mu_1$$

$$= C_{k,j_1} \mathbb{E}(|Q^2_{k,j} - Q^3_{k,j}|)_{\mu_1}$$

$$\tag{123}$$

$$\int_{\{(s,a),r',s'\}:A1 < 0, A2 < 0} (A1)(A2)d(s,a)d\mu_1(s,a)d\mu_2(r)d\mu_3(s^{'})d\mu_4(a_{i+1})$$

$$= C_{k,j_2} \int |Q^2_{k,j} - Q^3_{k,j}|d\nu$$

$$= C_{k,j_2} \mathbb{E}(|Q^2_{k,j} - Q^3_{k,j}|)_{\mu_1}$$

$$\tag{124}$$

We write the last two terms as

$$\int_{\{(s,a),r',s'\}:A1 \geq 0, A2 < 0} (A1)(A2)d\mu_1(s,a)d\mu_2(r)d\mu_3(s^{'})d\mu_4(a_{i+1}) = C_{k,j_3}\epsilon \tag{125}$$

$$\int_{\{(s,a),r',s'\}:A1 < 0, A2 \geq 0} (A1)(A2)d\mu_1(s,a)d\mu_2(r)d\mu_3(s^{'})d\mu_4(a_{i+1}) = C_{k,j_4}\epsilon \tag{126}$$

Here $C_{k,j_1}, C_{k,j_2}, C_{k,j_4}$ and $C_{k,j_4}$ are positive constants. Plugging Equations equation 123, equation 124, equation 125, equation 126 into Equation equation 121.

$$(C_{k,j_1} + C_{k,j_2})\mathbb{E}(|Q^2_{k,j} - Q^3_{k,j}|)_{\mu_1} - (C_{k,j_3} + C_{k,j_4})\epsilon \leq \epsilon \tag{127}$$

$$\tag{128}$$

which implies

$$\mathbb{E}(|Q_{k,j}^2 - Q_{k,j}^3|)_{\mu_1} \leq \left(\frac{1 + C_{k,j_3} + C_{k,j_4}}{C_{k,j_1} + C_{k,j_2}}\right)\epsilon \tag{129}$$

$$\tag{130}$$

Thus we have

$$\mathbb{E}(|Q_{k,j}^2 - Q_{k,j}^3|)_{\mu_1} \leq \left(\frac{1 + C_{k,j_3} + C_{k,j_4}}{C_{k,j_1} + C_{k,j_2}}\right)C_k\frac{1}{\sqrt{n}} \tag{131}$$

$$\tag{132}$$

which implies

$$\mathbb{E}(|Q_{k,j}^2 - Q_{k,j}^3|)_{\mu_1} \leq \tilde{\mathcal{O}}\left(\frac{1}{\sqrt{n}}\right) \tag{133}$$

$$\tag{134}$$

$$\square$$

### D.4 PROOF OF LEMMA 6

*Proof.* Note that from theorem 5 of Allen-Zhu et al. (2019) we have that for a neural network with at least $m$ neurons in every layer, sample size of $n$, we have after $T_{k,j}$ steps of the gradient descent algorithm with step size $\alpha$ we have with probability at least $1 - \exp(-\Omega(\log(m)))$

$$F(\theta^T) \leq \left(1 - \Omega\left(\frac{\alpha m}{n^2}\right)\right)^{T_{k,j}} F(\theta^0) \tag{135}$$

Where $F$ is the loss function to be minimized. The specific form of $F(\theta^T)$ is obtained in the proof of theorem 1 in the appendix. Note that we have ignored the minimum euclidean distance between training samples denoted as $\delta$ in Allen-Zhu et al. (2019). It is to be noted that this is non-zero for our case as for any two separate $s, a$ pairs the target function will be different for a non-zero neural network.

Now plug gin in $m \geq \mathcal{O}(\delta^{-1})$ we obtain with probability at least $1 - \delta$

$$F(\theta^{T_{k,j}}) \leq \mathcal{O}\left(1 - \Omega\left(\frac{\alpha m}{n^2}\right)\right)^{T_{k,j}} \tag{136}$$

Now choose a constant $C'_{k,j}$ such that

$$C'_{k,j}|\theta^{T_{k,j}} - \theta^*| \leq F(\theta^T) \leq \mathcal{O}\left(1 - \Omega\left(\frac{\alpha m}{n^2}\right)\right)^{T_{k,j}} \tag{137}$$

where $\theta^*$ is the global minima for the loss function $F$. Now using the Lipschitz property of neural networks with respect to the parameters as discussed in Reddi et al. (2019) we have

$$|Q_{\theta^T}(s,a) - Q_{\theta^*}(s,a)| \leq (L_{k,j})(C'_{k,j})(\theta^T - \theta^*) \leq (L_{k,j})F(\theta^T) \leq \mathcal{O}\left(1 - \Omega\left(\frac{\alpha m}{n^2}\right)\right)^{T_{k,j}} \tag{138}$$

for all $(s,a) \in \mathcal{S} \times \mathcal{A}$. Note that by definition $Q_{\theta^T}(s,a)$ is $Q_{k,j}$ and $Q_{\theta^*}(s,a)$ is $Q_{k,j}^3$, thus we have with probability at least $1 - \delta$

$$|Q_{\theta^T}(s,a) - Q_{\theta^*}(s,a)| \leq \mathcal{O}\left(1 - \Omega\left(\frac{\alpha m}{n^2}\right)\right)^{T_{k,j}} \tag{139}$$

Now taking expectation on both side with respect to $(s,a) \sim \zeta_{\pi^{\lambda_k}}^{\nu}$ gives us the required result. $\square$

### D.5 PROOF OF LEMMA 7

*Proof.* Note that this lemma is a direct application of theorem 1 from Doan (2022). Note that in the second inner for loop of Algorithm 1 we perform an iterations for every state action pair sampled in the first inner for loop for a total of $J \cdot n$ samples and noting that we are taking a steps of gradient descent for each sample gives us the required result. □

## E COMPARISON OF SAMPLE COMPLEXITY ANALYSIS WITH NATURAL POLICY GRADIENT

For natural policy gradient (NPG) (Agarwal et al., 2021), to derive the sample complexity result, the average error in estimation till iteration $K$ is given by

$$\underset{k \in \{1, \cdots, K\}}{\arg\min} V^*(\nu) - V^{\pi_K}(\nu) \leq \left( \frac{\log(|\mathcal{A}|)}{K\eta(1-\gamma)} + \frac{\eta\beta_k W^2}{2(1-\gamma)} + \frac{1}{K} \sum_{i=k}^{K} \frac{err_k}{1-\gamma} \right), \quad (140)$$

where $err_k$ in the last term on the right-hand side of equation 140 is

$$err_k = \mathbb{E}_{s,a}(A^{\pi_{\lambda_K}} - w^k \nabla log(\pi_{\lambda_k}(a|s))) \quad (141)$$

where $s \sim d_\nu^{\pi^*}, a \sim \pi^*(.|s)$, $w_k$ is our estimate of the NPG gradient update term and $\lambda_k$ is the policy parameter.

The term $err_k$ is then decomposed in the following manner

$$\mathbb{E}_{s,a}(A^{\pi_{\lambda_K}} - w^k \nabla log(\pi_{\lambda_k}(a|s))) = \mathbb{E}_{s,a}(A^{\pi_{\lambda_K}} - w^* \nabla log(\pi_{\lambda_k}(a|s)))$$
$$+ (w^* - w^k) \nabla log(\pi_{\lambda_k}(a|s))) \quad (142)$$

where $w^* = \arg\min_w \mathbb{E}_{s,a}(Q^{\pi_{\lambda_K}} - w \nabla log(\pi_{\lambda_k}(a|s)))$ where $s \sim d_\nu^{\pi_{\lambda_k}}, a \sim \pi^{\lambda_k}(.|s)$.

For ease of notation we define

$$\mathbb{E}_{s \sim d_\nu^\pi, a \sim \pi(.|s)}(Q^{\pi_\lambda} - w \nabla log(\pi_\lambda(a|s)))^2 = L(w, \lambda, d_\nu^\pi). \quad (143)$$

Equation equation 142 is then be upper bounded as

$$\mathbb{E}_{s,a}(Q^{\pi_{\lambda_K}} - w^k \nabla log(\pi_{\lambda_k}(a|s))) \leq \sqrt{L(w^*, \lambda_k, d_\nu^{\pi^*})}$$
$$+ \phi_k \sqrt{L(w^k, \lambda_k, d_\nu^{\pi_{\lambda_k}}) - L(w^*, \lambda_k, d_\nu^{\pi_{\lambda_k}})} \quad (144)$$

where $\phi_k$ is a constant which represents the change in expectation from $d_\nu^{\pi^*}$ to $d_\nu^{\pi_{\lambda_k}}$.

Assumption 6.1 and 6.2 in (Agarwal et al., 2021) are as follows

$$L(w^k, \lambda_k, d_\nu^{\pi_{\lambda_k}}) - L(w^*, \lambda_k, d_\nu^{\pi_{\lambda_k}}) \leq \epsilon_{stat} \quad (145)$$
$$L(w^*; \lambda_k, d_\nu^{\pi_\nu}) \leq \epsilon_{bias} \quad (146)$$

$\forall k \in \{1, \cdots, K\}$, where $K$ is the total number of iterations of the NPG algorithm.

The assumption in Equation equation 145 is known as the excess risk assumption and places an upper bound on the error incurred due to the difference between the obtained estimate $w^k$ and the optimal solution $w^*$ which minimizes $L(w, \lambda_k, d_\nu^{\pi_{\lambda_k}})$. It is a measure of uncertainty in estimating the natural gradient update.

The assumption in Equation equation 146 is known as the transfer error assumption and places an upper bound on the loss function $L(w, \lambda_k, d_\nu^{\pi^*})$ evaluated at the minima of the loss function $L(w; \lambda^k, d_\nu^{\pi_{\lambda_k}})$. This is a measure of how similar the policy $\pi^{\lambda_k}$ is to the optimal policy $\pi^*$.

In the analysis of Agarwal et al. (2021), using results of stochastic gradient descent on a convex loss function, $\epsilon_{stat}$ is assumed to be upper bounded as $\tilde{\mathcal{O}}\left(\frac{1}{\sqrt{n_k}}\right)$ where $n_k$ is the number of state action samples at iteration $k$. Further, $\epsilon_{bias}$ is directly assumed as a constant while it depends on the accurate estimation of $A^{\pi_{\lambda_k}}$.

**Comparison.** We note that the analysis in Equation equation 145-equation 146 does not consider (i) the extra $\left(\frac{1}{1-\gamma}\right)$ state action samples required to obtain Monte Carlo estimate $A^{\pi_{\lambda_k}}$. This is because each Monte Carlo estimate of $A^{\pi_{\lambda_k}}$ requires on average $\left(\frac{1}{1-\gamma}\right)$ state action samples; (ii) the error incurred due to gap between the Monte Carlo estimate $A^{\pi_{\lambda_k}}$ and the actual Q-function. In Agarwal et al. (2021), Monte Carlo estimate is only shown to be an unbiased estimate of $A^{\pi_{\lambda_k}}$ and no error bound for the estimate is given. This error bound will require additional samples to be very close such that the obtained value function for the policy is $\epsilon$-close. This is the key gap due to which our algorithm gets additional $1/\epsilon$ in the sample complexity.

Our analysis considers the number of samples required to estimate $A^{\pi_{\lambda_k}}$ to a given accuracy in our sample complexity analysis. In order to account for the difference between the optimal policy $\pi^*$ and the policy estimate $\pi_{\lambda_k}$, which has been used and verified in prior works such as Farahmand et al. (2010).

The authors in (Liu et al., 2020b) also perform a similar analysis but only has an Assumption similar to Equation equation 146. This assumption also suffers from the same drawback described above.

## F  SAMPLE COMPLEXITY USING CONVEX REFORMULATION WITH TWO-LAYER NEURAL NETWORKS

We now demonstrate how the error bound in Theorem 1 can be made deterministic if we use a 2 layer neural network with ReLU activation functions.

This part of the Appendix will first go into detail as to how a 2 layer neural network can be reformulated as a convex problem. We then prove the supplementary lemmas and additional assumptions required to prove a deterministic version of Theorem 1.

A 2-layer ReLU Neural Network with input $x \in \mathbb{R}^d$ is defined as $f(x) = \sum_{i=1}^m \sigma'(x^T u_i)\alpha_i$, where $m \geq 1$ is the number of neurons in the neural network, the parameter space is $\Theta_m = \mathbb{R}^{d \times m} \times \mathbb{R}^m$ and $\theta = (U, \alpha)$ is an element of the parameter space, where $u_i$ is the $i^{th}$ column of $U$, and $\alpha_i$ is the $i^{th}$ coefficient of $\alpha$. The function $\sigma' : \mathbb{R} \to \mathbb{R}_{\geq 0}$ is the ReLU or restricted linear function defined as $\sigma'(x) \triangleq \max(x, 0)$. In order to obtain parameter $\theta$ for a given set of data $X \in \mathbb{R}^{n \times d}$ and the corresponding response values $y \in \mathbb{R}^{n \times 1}$, we desire the parameter that minimizes the squared loss, given by

$$\mathcal{L}(\theta) = \left\| \sum_{i=1}^m \sigma(Xu_i)\alpha_i - y \right\|_2^2. \tag{147}$$

In equation 147, we have the term $\sigma(Xu_i)$ which is a vector $\{\sigma'((x_j)^T u_i)\}_{j \in \{1, \cdots, n\}}$, where $x_j$ is the $j^{th}$ row of $X$. It is the ReLU function applied to each element of the vector $Xu_i$. We note that the optimization in Equation equation 147 is non-convex in $\theta$ due to the presence of the ReLU activation function. In Wang et al. (2021b), it is shown that this optimization problem has an equivalent convex form, provided that the number of neurons $m$ goes above a certain threshold value. This convex problem is obtained by replacing the ReLU functions in the optimization problem with equivalent diagonal operators. The convex problem is given as

$$\mathcal{L}'_\beta(p) := \left\| \sum_{D_i \in D_X} D_i(Xp_i) - y \right\|_2^2, \tag{148}$$

where $p \in \mathbb{R}^{d \times |D_X|}$. $D_X$ is the set of diagonal matrices $D_i$ which depend on the data-set $X$. Except for cases of $X$ being low rank, it is not computationally feasible to obtain the set $D_X$. We instead use $\tilde{D} \in D_X$ to solve the convex problem in equation 148 where $p$ now would lie in $p \in \mathbb{R}^{d \times |\tilde{D}|}$.

For a set of parameters $\theta = (u, \alpha) \in \Theta$, we denote neural network represented by these parameters as

$$Q_\theta(s, a) = \sum_{i=1}^{m} \sigma'((s, a)^T u_i) \alpha_i. \tag{149}$$

For representing the action value function, we will use a 2 layer ReLU neural network. In this section, we first lay out the theory behind the convex formulation of the 2 layer ReLU neural network. In the next section it will shown how it is utilised for the FQI algorithm.

In order to obtain parameter $\theta$ for a given set of data $X \in \mathbb{R}^{n \times d}$ and the corresponding response values $y \in \mathbb{R}^{n \times 1}$, we desire the parameter that minimizes the squared loss (with a regularization parameter $\beta \in [0, 1]$), given by

$$\mathcal{L}(\theta) \quad = \quad \left\| \sum_{i=1}^{m} \sigma(X u_i) \alpha_i - y \right\|_2^2. \tag{150}$$

Here, we have the term $\sigma(X u_i)$ which is a vector $\{\sigma'((x_j)^T u_i)\}_{j \in \{1, \cdots, n\}}$ where $x_j$ is the $j^{th}$ row of $X$. It is the ReLU function applied to each element of the vector $X u_i$. We note that the optimization in Equation equation 147 is non-convex in $\theta$ due to the presence of the ReLU activation function. In Wang et al. (2021b), it is shown that this optimization problem has an equivalent convex form, provided that the number of neurons $m$ goes above a certain threshold value. This convex problem is obtained by replacing the ReLU functions in the optimization problem with equivalent diagonal operators. The convex problem is given as

$$\mathcal{L}'_\beta(p) \quad := \quad \left\| \sum_{D_i \in D_X} D_i(X p_i) - y \right\|_2^2 \tag{151}$$

where $p \in \mathbb{R}^{d \times |D_X|}$. $D_X$ is the set of diagonal matrices $D_i$ which depend on the data-set $X$. Except for cases of $X$ being low rank it is not computationally feasible to obtain the set $D_X$. We instead use $\tilde{D} \in D_X$ to solve the convex problem

$$\mathcal{L}'_\beta(p) \quad := \quad \left\| \sum_{D_i \in \tilde{D}} D_i(X p_i) - y \right\|_2^2, \tag{152}$$

where $p \in \mathbb{R}^{d \times |\tilde{D}|}$. In order to understand the convex reformulation of the squared loss optimization problem, consider the vector $\sigma(X u_i)$

$$\sigma(X u_i) = \begin{bmatrix} \{\sigma'((x_1)^T u_i)\} \\ \{\sigma'((x_2)^T u_i)\} \\ \vdots \\ \{\sigma'((x_n)^T u_i)\}. \end{bmatrix} \tag{153}$$

Now for a fixed $X \in \mathbb{R}^{n \times d}$, different $u_i \in \mathbb{R}^{d \times 1}$ will have different components of $\sigma(X u_i)$ that are non zero. For example, if we take the set of all $u_i$ such that only the first element of $\sigma(X u_i)$ are non zero (i.e, only $(x_1)^T u_i \geq 0$ and $(x_j)^T u_i < 0 \; \forall j \in [2, \cdots, n]$ ) and denote it by the set $\mathcal{K}_1$, then we have

$$\sigma(X u_i) = D_1(X u_i) \quad \forall u_i \in \mathcal{K}_1, \tag{154}$$

where $D_1$ is the $n \times n$ diagonal matrix with only the first diagonal element equal to 1 and the rest 0. Similarly, there exist a set of $u's$ which result in $\sigma(X u)$ having certain components to be non-zero and the rest zero. For each such combination of zero and non-zero components, we will have a corresponding set of $u_i's$ and a corresponding $n \times n$ Diagonal matrix $D_i$. We define the possible set of such diagonal matrices possible for a given matrix X as

$$D_X = \{D = diag(\mathbf{1}(X u \geq 0)) : u \in \mathbb{R}^d, D \in \mathbb{R}^{n \times n}\}, \tag{155}$$

where $diag(\mathbf{1}(Xu \geq 0))$ represents a matrix given by

$$D_{k,j} = \begin{cases} \mathbf{1}(x_j^T u), & \text{for } k = j \\ 0 & \text{for } k \neq j \end{cases}, \tag{156}$$

where $\mathbf{1}(x) = 1$ if $x > 0$ and $\mathbf{1}(x) = 0$ if $x \leq 0$. Corresponding to each such matrix $D_i$, there exists a set of $u_i$ given by

$$\mathcal{K}_i = \{u \in \mathbb{R}^d : \sigma(Xu_i) = D_i Xu_i, D_i \in D_X\} \tag{157}$$

where $I$ is the $n \times n$ identity matrix. The number of these matrices $D_i$ is upper bounded by $2^n$. From Wang et al. (2021b) the upper bound is $\mathcal{O}\left(r\left(\frac{n}{r}\right)^r\right)$ where $r = rank(X)$. Also, note that the sets $\mathcal{K}_i$ form a partition of the space $\mathbb{R}^{d \times 1}$. Using these definitions, we define the equivalent convex problem to the one in Equation equation 147 as

$$\mathcal{L}_\beta(v, w) := \left(\left\|\sum_{D_i \in D_X} D_i(X(v_i - w_i)) - y\right\|_2^2\right), \tag{158}$$

where $v = \{v_i\}_{i \in 1, \cdots, |D_X|}, w = \{w_i\}_{i \in 1, \cdots, |D_X|}, v_i, w_i \in \mathcal{K}_i$, note that by definition, for any fixed $i \in \{1, \cdots, |D_X|\}$ at-least one of $v_i$ or $w_i$ are zero. If $v^*, w^*$ are the optimal solutions to Equation equation 158, the number of neurons $m$ of the original problem in Equation equation 147 should be greater than the number of elements of $v^*, w^*$, which have at-least one of $v_i^*$ or $w_i^*$ non-zero. We denote this value as $m_{X,y}^*$, with the subscript $X$ denoting that this quantity depends upon the data matrix $X$ and response $y$.

We convert $v^*, w^*$ to optimal values of Equation equation 147, denoted by $\theta^* = (U^*, \alpha^*)$, using a function $\psi : \mathbb{R}^d \times \mathbb{R}^d \to \mathbb{R}^d \times \mathbb{R}$ defined as follows

$$\psi(v_i, w_i) = \begin{cases} (v_i, 1), & \text{if } w_i = 0 \\ (w_i, -1), & \text{if } v_i = 0 \\ (0, 0), & \text{if } v_i = w_i = 0 \end{cases} \tag{159}$$

where according to Pilanci & Ergen (2020) we have $(u_i^*, \alpha_i^*) = \psi(v_i^*, w_i^*)$, for all $i \in \{1, \cdots, |D_X|\}$ where $u_i^*, \alpha_i^*$ are the elements of $\theta^*$. Note that restriction of $\alpha_i$ to $\{1, -1, 0\}$ is shown to be valid in Mishkin et al. (2022). For $i \in \{|D_X| + 1, \cdots, m\}$ we set $(u_i^*, \alpha_i^*) = (0, 0)$.

Since $D_X$ is hard to obtain computationally unless $X$ is of low rank, we can construct a subset $\tilde{D} \in D_X$ and perform the optimization in Equation equation 158 by replacing $D_X$ with $\tilde{D}$ to get

$$\mathcal{L}_\beta(v, w) := \left(\left\|\sum_{D_i \in \tilde{D}} D_i(X(v_i - w_i)) - y\right\|_2^2\right) \tag{160}$$

where $v = \{v_i\}_{i \in 1, \cdots, |\tilde{D}|}, w = \{w_i\}_{i \in 1, \cdots, |\tilde{D}|}, v_i, w_i \in \mathcal{K}_i$, by definition, for any fixed $i \in \{1, \cdots, |\tilde{D}|\}$ at-least one of $v_i$ or $w_i$ are zero.

The required condition for $\tilde{D}$ to be a sufficient replacement for $D_X$ is as follows. Suppose $(v, w) = (\bar{v}_i, \bar{w}_i)_{i \in (1, \cdots, |\tilde{D}|)}$ denote the optimal solutions of Equation equation 160. Then we require

$$m \geq \sum_{D_i \in \tilde{D}} |\{\bar{v}_i : \bar{v}_i \neq 0\} \cup \{\bar{w}_i : \bar{w}_i \neq 0\}|. \tag{161}$$

Or, the number of neurons in the neural network are greater than the number of indices $i$ for which at-least one of $v_i^*$ or $w_i^*$ is non-zero. Further,

$$diag(Xu_i^* \geq 0 : i \in [m]) \in \tilde{D}. \tag{162}$$

In other words, the diagonal matrices induced by the optimal $u_i^*$'s of Equation equation 147 must be included in our sample of diagonal matrices. This is proved in Theorem 2.1 of Mishkin et al. (2022).

A computationally efficient method for obtaining $\tilde{D}$ and obtaining the optimal values of the Equation equation 147, is laid out in Mishkin et al. (2022). In this method we first get our sample of diagonal matrices $\tilde{D}$ by first sampling a fixed number of vectors from a $d$ dimensional standard multivariate

distribution, multiplying the vectors with the data matrix $X$ and then forming the diagonal matrices based of which co-ordinates are positive. Then we solve an optimization similar to the one in Equation equation 158, without the constraints, that its parameters belong to sets of the form $\mathcal{K}_i$ as follows.

$$\mathcal{L}'_\beta(p) := \left( \left\| \sum_{D_i \in \tilde{D}} D_i(Xp_i) - y \right\|_2^2 \right), \tag{163}$$

where $p \in \mathbb{R}^{d \times |\tilde{D}|}$ . In order to satisfy the constraints of the form given in Equation equation 158, this step is followed by a cone decomposition step. This is implemented through a function $\{\psi'_i\}_{i \in \{1, \cdots, |\tilde{D}|\}}$. Let $p^* = \{p_i^*\}_{i \in \{1, \cdots, |\tilde{D}|\}}$ be the optimal solution of Equation equation 163. For each $i$ we define a function $\psi'_i : \mathbb{R}^d \to \mathbb{R}^d \times \mathbb{R}^d$ as

$$\begin{aligned} \psi'_i(p_i) &= (v_i, w_i) \\ \textit{such that } p &= v_i - w_i, \textit{and } v_i, w_i \in \mathcal{K}_i \end{aligned} \tag{164}$$

Then we obtain $\psi(p_i^*) = (\bar{v}_i, \bar{w}_i)$. As before, at-least one of $v_i$, $w_i$ is 0. Note that in practice we do not know if the conditions in Equation equation 161 and equation 162 are satisfied for a given sampled $\tilde{D}$. We express this as follows. If $\tilde{D}$ was the full set of Diagonal matrices then we would have $(\bar{v}_i, \bar{w}_i) = v_i^*, w_i^*$ and $\psi(\bar{v}_i, \bar{w}_i) = (u_i^*, \alpha_i^*)$ for all $i \in (1, \cdots, |D_X|)$. However, since that is not the case and $\tilde{D} \in D_X$, this means that $\{\psi(\bar{v}_i, \bar{w}_i)\}_{i \in (1, \cdots, |\tilde{D}|)}$ is an optimal solution of a non-convex optimization different from the one in Equation equation 147. We denote this non-convex optimization as $\mathcal{L}_{|\tilde{D}|}(\theta)$ defined as

$$\mathcal{L}_{|\tilde{D}|}(\theta) = \left\| \sum_{i=1}^{m'} \sigma(Xu_i)\alpha_i - y \right\|_2^2, \tag{165}$$

where $m' = |\tilde{D}|$ or the size of the sampled diagonal matrix set. In order to quantify the error incurred due to taking a subset of $D_X$, we assume that the expectation of the absolute value of the difference between the neural networks corresponding to the optimal solutions of the non-convex optimizations given in Equations equation 165 and equation 147 is upper bounded by a constant depending on the size of $\tilde{D}$. The formal assumption and its justification is given in Assumption 5.

## F.1 PROPOSED NATURAL ACTOR CRITIC ALGORITHM WITH 2-LAYER CRITIC PARAMETRIZATION (NAC2L)

We summarize the proposed approach in Algorithm 2. Algorithm 2 has an outer for loop with two inner for loops. At a fixed iteration $k$ of the outer for loop and iteration $j$ of the first inner for loop, we obtain a sequence of state action pairs and the corresponding state and reward by following the estimate of the policy at the start of the iteration. In order to perform the critic update, the state action pairs and the corresponding target $Q$ values are stored in matrix form and passed to Algorithm 3, as the input and output values respectively to solve the following optimization problem.

$$\arg\min_{\theta \in \Theta} \frac{1}{n} \sum_{i=1}^n \left( Q_\theta(s_i, a_i) - r(s_i, a_i) - \gamma Q_{k,j-1}(s_{i+1}, a_{i+1}) \right)^2, \tag{167}$$

where $Q_{k,j-1}$ is the estimate of the $Q$ function at the $k^{th}$ iteration of the outer for loop and the $(j-1)^{th}$ iteration of the first inner for loop of Algorithm 2. $Q_\theta$ is a neural network defined as in equation 149 and $n$ is the number of state action pairs sampled at the $k^{th}$ iteration of the outer for loop and the $j^{th}$ iteration of the first inner for loop of Algorithm 2. This is done at each iteration of the first inner for loop to perform what is known as a Fitted Q-iteration step to obtain the estimate of the critic.

Algorithm 3 first samples a set of diagonal matrices denoted by $\tilde{D}$ in line 2 of Algorithm 3. The elements of $\tilde{D}$ act as the diagonal matrix replacement of the ReLU function. Algorithm 3 then solves an optimization of the form given in Equation equation 167 by converting it to an optimization of the form equation 152. This convex optimization is solved in Algorithm 3 using the projected

---

**Algorithm 2** Natural Actor Critic with 2-Layer Critic Parametrization (NAC2L)

---

**Input: Input:** $\mathcal{S}, \mathcal{A}, \gamma$, Time Horizon $K \in \mathcal{Z}$ , Updates per time step $J \in \mathcal{Z}$ ,starting state sampling distribution $\nu$, Actor step sizes $\beta_{i,k}, \forall k \in \{1, \cdots, K\}, i \in \{1, \cdots, n.J\}$, Critic step size $\alpha$, policy gradient step size $\eta$, Number of convex optimization steps $T_{k,j}, k \in \{1, \cdots, K\}, j \in \{1, \cdots, J\}$,

1: **Initialize:** $\lambda_0 = \{0\}^d$
2: **for** $k \in \{1, \cdots, K\}$ **do**
3:      $X_k = \varnothing$
4:      **for** $j \in \{1, \cdots, J\}$ **do**
5:          Take $n$ state action pairs sampled from $\nu$ as the starting state distribution and then following policy $\pi_{\lambda_k}$.
6:          Set $y_i = r_i + \gamma Q_{k,j-1}(s_{i+1}, a_{i+1})$, where $i \in \{1, \cdots, n\}$
7:          Set $X_j, Y_j$ as the matrix of the sampled state action pairs and vector of estimated $Q$ values respectively
8:          $X_k = X_k \cup X_j$
9:          **Call Algorithm** 3 with input $(X = X_j, y = Y_j, T = T_{k,j})$ and return parameter $\theta$
10:        $Q_{k,j} = Q_\theta$
11:      **end for**
12:      $w_0 = 0^d$
13:      **for** $i \in \{1, \cdots, |X_k|\}$ **do**
14:          $A_{k,J}(s_i, a_i) = Q_{k,J}(s_i, a_i) - \sum_{a \in \mathcal{A}} \pi_{\lambda_k}(a|s_i) Q_{k,J}(s_i, a)$
15:          $w_i = w_i - \beta_i \Big( w_i \cdot \nabla_\lambda \log \pi_{\lambda_k}(a_i|s_i) - A_{k,J}(s_i, a_i) \Big) \nabla_\lambda \log \pi_{\lambda_k}(a_i|s_i)$
16:      **end for**
17:      Update $\lambda_{k+1} = \lambda_k + \eta \left( \frac{1}{1-\gamma} \right) w_{|X_k|}$
18: **end for**
     Output: $\pi_{\lambda_{K+1}}$

---

**Algorithm 3** Neural Network Parameter Estimation

---

1: **Input:** data $(X, y, T)$
2: **Sample:** $\tilde{D} = diag(1(Xg_i > 0)) : g_i \sim \mathcal{N}(0, I), i \in [|\tilde{D}|]$
3: **Initialize** $y^1 = 0, u^1 = 0$
     **Initialize** $g(u) = \| \sum_{D_i \in \tilde{D}} D_i X u_i - y \|_2^2$
4: **for** $i \in \{0, \cdots, T\}$ **do**
5:      $u^{i+1} = y_i - \alpha_i \nabla g(y_i)$
6:      $y^{i+1} = \arg \min_{y:|y|_1 \leq \frac{R_{max}}{1-\gamma}} \|u_{i+1} - y\|_2^2$
7: **end for**
8: Set $u^{T+1} = u^*$
9: Solve Cone Decomposition:
     $\bar{v}, \bar{w} \in u_i^* = v_i - w_i, i \in [d]\}$ such that $v_i, w_i \in \mathcal{K}_i$ and at-least one $v_i, w_i$ is zero.
10: Construct $(\theta = \{u_i, \alpha_i\})$ using the transformation

$$\psi(v_i, w_i) = \begin{cases} (v_i, 1), & \text{if } w_i = 0 \\ (w_i, -1), & \text{if } v_i = 0 \\ (0, 0), & \text{if } v_i = w_i = 0 \end{cases} \tag{166}$$

     for all $i \in \{1, \cdots, m\}$
11: Return $\theta$

---

gradient descent algorithm. After obtaining the optima for this convex program, denoted by $u^* = \{u_i^*\}_{i \in \{1, \cdots, |\tilde{D}|\}}$, in line 10, we transform them into an estimate of the solutions for the optimization given in equation 167, which are then passed back to Algorithm 2. The procedure is described in detail along with the relevant definitions in Appendix F.

The estimate of $w_k^*$ is obtained in the second inner for loop of Algorithm equation 2 where a gradient descent is performed for the loss function of the form given in Equation equation 9 using the state

action pairs sampled in the first inner for loop. Note that we do not have access to the true $Q$ function that is required for the critic update. Thus we use the estimate of the $Q$ function obtained at the end of the first inner for loop. After obtaining our estimate of the minimizer of Equation equation 9, we update the policy parameter using the stochastic gradient update step. Here the state action pairs used are the same we sampled in the first inner for loop.

Now since we do not know the exact convex reformulation but instead use an approximation, we have the following additional assumption.

**Assumption 5.** *Let $\theta^* \triangleq \arg\min_{\theta \in \Theta} \mathcal{L}(\theta)$, where $\mathcal{L}(\theta)$ is defined in equation 147 and we denote $Q_{\theta^*}(\cdot)$ as $Q_\theta(\cdot)$ as defined in equation 149 for $\theta = \theta^*$. Also, let $\theta^*_{\tilde{D}} \triangleq \arg\min_{\theta \in \Theta} \mathcal{L}_{|\tilde{D}|}(\theta)$, where $\mathcal{L}_{\tilde{D}}(\theta)$ is the loss function $\mathcal{L}(\theta)$ with the set of diagonal matrices $D$ replaced by $\tilde{D} \in D$. Further, we denote $Q_{\theta^*_{|\tilde{D}|}}(\cdot)$ as $Q_\theta(\cdot)$ as defined in equation 149 for $\theta = \theta^*_{|\tilde{D}|}$. Then we assume*

$$\mathbb{E}_{s,a}(|Q_{\theta^*} - Q_{\theta^*_{|\tilde{D}|}}|)_\nu \leq \epsilon_{|\tilde{D}|}, \tag{168}$$

*for any $(s,a) \sim \zeta^\nu_{\pi^{\lambda_k}}$.*

Thus, $\epsilon_{|\tilde{D}|}$ is a measure of the error incurred due to taking a sample of diagonal matrices $\tilde{D}$ and not the full set $D_X$. In practice, setting $|\tilde{D}|$ to be the same order of magnitude as $d$ (dimension of the data) gives us a sufficient number of diagonal matrices to get a reformulation of the non convex optimization problem which performs comparably or better than existing gradient descent algorithms, therefore $\epsilon_{|\tilde{D}|}$ is only included for theoretical completeness and will be negligible in practice. This has been practically demonstrated in Mishkin et al. (2022); Bartan & Pilanci (2022); Sahiner et al. (2022).

Before proceeding with the proof further, we would like to prove two supplementary lemmas

**Lemma 8.** *Consider an optimization of the form given in Equation equation 160 denoted by $\mathcal{L}_{|\tilde{D}|}$ and it's convex equivalent denoted by $\mathcal{L}_0$. Then the value of these two loss functions evaluated at $(v,w) = (v_i, w_i)_{i \in \{1, \cdots, |\tilde{D}|\}}$ and $\theta = \psi(v_i, w_i)_{i \in \{1, \cdots, |\tilde{D}|\}}$ respectively are equal and thus we have*

$$\mathcal{L}_{|\tilde{D}|}(\psi(v_i, w_i)_{i \in \{1, \cdots, |\tilde{D}|\}}) = \mathcal{L}_0((v_i, w_i)_{i \in \{1, \cdots, |\tilde{D}|\}}) \tag{169}$$

*Proof.* Consider the loss functions in Equations equation 158, equation 163 are as follows

$$\mathcal{L}_0((v_i, w_i)_{i \in \{1, \cdots, |\tilde{D}|\}}) = || \sum_{D_i \in \tilde{D}} D_i(X(v_i - w_i)) - y ||_2^2 \tag{170}$$

$$\mathcal{L}_{|\tilde{D}|}(\psi(v_i, w_i)_{i \in \{1, \cdots, |\tilde{D}|\}}) = || \sum_{i=1}^{|\tilde{D}|} \sigma(X\psi(v_i, w_i)_1)\psi(v_i, w_i)_2 - y ||_2^2, \tag{171}$$

where $\psi(v_i, w_i)_1, \psi(v_i, w_i)_2$ represent the first and second coordinates of $\psi(v_i, w_i)$ respectively.

For any fixed $i \in \{1, \cdots, |\tilde{D}|\}$ consider the two terms

$$D_i(X(v_i - w_i)) \tag{172}$$

$$\sigma(X\psi(v_i, w_i)_1)\psi(v_i, w_i)_2 \tag{173}$$

For a fixed $i$ either $v_i$ or $w_i$ is zero. In case both are zero, both of the terms in Equations equation 172 and equation 173 are zero as $\psi(0,0) = (0,0)$. Assume that for a given $i$ $w_i = 0$. Then we have $\psi(v_i, w_i) = (v_i, 1)$. Then equations equation 172, equation 173 are.

$$D_i(X(v_i) \tag{174}$$

$$\sigma(X(v_i)) \tag{175}$$

But by definition of $v_i$ we have $D_i(X(v_i) = \sigma(X(v_i))$, therefore Equations equation 174, equation 175 are equal. Alternatively if for a given $i$ $v_i = 0$, then $\psi(v_i, w_i) = (w_i, -1)$, then the terms in equation 172, equation 173 become.

$$-D_i(X(w_i) \tag{176}$$
$$-\sigma(X(w_i)) \tag{177}$$

By definition of $w_i$ we have $D_i(X(w_i) = \sigma(X(w_i))$, then the terms in equation 176, equation 176 are equal. Since this is true for all $i$, we have

$$\mathcal{L}_{|\tilde{D}|}(\psi(v_i, w_i)_{i \in \{1, \cdots, |\tilde{D}|\}}) = \mathcal{L}_0((v_i, w_i)_{i \in \{1, \cdots, |\tilde{D}|\}}) \tag{178}$$

$\square$

**Lemma 9.** *The function $Q_\theta(x)$ defined in equation equation 149 is Lipschitz continuous in $\theta$, where $\theta$ is considered a vector in $\mathbb{R}^{(d+1)m}$ with the assumption that the set of all possible $\theta$ belong to the set $\mathcal{B} = \{\theta : |\theta^* - \theta|_1 < 1\}$, where $\theta^*$ is some fixed value.*

*Proof.* First we show that for all $\theta_1 = \{u_i, \alpha_i\}, \theta_2 = \{u_i', \alpha_i'\} \in \mathcal{B}$ we have $\alpha_i = \alpha_i'$ for all $i \in (1, \cdots, m)$

Note that

$$|\theta_1 - \theta_2|_1 = \sum_{i=1}^m |u_i - u_i'|_1 + \sum_{i=1}^m |\alpha_i - \alpha_i'|, \tag{179}$$

where $|u_i - u_i'|_1 = \sum_{j=1}^d |u_{i_j} - u_{i_j}'|$ with $u_{i_j}, u_{i_j}'$ denote the $j^{th}$ component of $u_i, u_i'$ respectively.

By construction $\alpha_i, \alpha_i'$ can only be 1, $-1$ or 0. Therefore if $\alpha_i \neq \alpha_i'$ then $|\alpha_i - \alpha_i'| = 2$ if both non zero or $|\alpha_i - \alpha_i'| = 1$ if one is zero. Therefore $|\theta_1 - \theta_2|_1 \geq 1$. Which leads to a contradiction.

Therefore $\alpha_i = \alpha_i'$ for all $i$ and we also have

$$|\theta_1 - \theta_2|_1 = \sum_{i=1}^m |u_i - u_i'|_1 \tag{180}$$

$Q_\theta(x)$ is defined as

$$Q_\theta(x) = \sum_{i=1}^m \sigma'(x^T u_i)\alpha_i \tag{181}$$

From Proposition 1 in Scaman & Virmaux (2018) the function $Q_\theta(x)$ is Lipschitz continuous in $x$, therefore there exist $l > 0$ such that

$$|Q_\theta(x) - Q_\theta(y)| \leq l|x - y|_1 \tag{182}$$
$$|\sum_{i=1}^m \sigma'(x^T u_i)\alpha_i - \sum_{i=1}^m \sigma'(y^T u_i)\alpha_i| \leq l|x - y|_1 \tag{183}$$

If we consider a single neuron of $Q_\theta$, for example $i = 1$, we have $l_1 > 0$ such that

$$|\sigma'(x^T u_1)\alpha_i - \sigma'(y^T u_1)\alpha_i| \leq l_1|x - y|_1 \tag{184}$$

Now consider Equation equation 184, but instead of considering the left hand side a a function of $x, y$ consider it a function of $u$ where we consider the difference between $\sigma'(x^T u)\alpha_i$ evaluated at $u_1$ and $u_1'$ such that

$$|\sigma'(x^T u_1)\alpha_i - \sigma'(x^T u_1')\alpha_i| \leq l_1^x |u_1 - u_1'|_1 \tag{185}$$

for some $l_1^x > 0$.

Similarly, for all other $i$ if we change $u_i$ to $u_i'$ to be unchanged we have

$$|\sigma'(x^T u_i)\alpha_i - \sigma'(x^T u_i')\alpha_i| \leq l_i^x |u_i - u_i'|_1 \tag{186}$$

for all $x$ if both $\theta_1, \theta_2 \in \mathcal{B}$.

Therefore we obtain

$$|\sum_{i=1}^m \sigma'(x^T u_i)\alpha_i - \sum_{i=1}^m \sigma'(x^T u_i')\alpha_i| \leq \sum_{i=1}^m |\sigma'(x^T u_i)\alpha_i - (x^T u_i')\alpha_i| \tag{187}$$

$$\leq \sum_{i=1}^m l_i^x |u_i - u_i'|_1 \tag{188}$$

$$\leq (\sup_i l_i^x) \sum_{i=1}^m |u_i - u_i'|_1 \tag{189}$$

$$\leq (\sup_i l_i^x)|\theta_1 - \theta_2| \tag{190}$$

This result for a fixed $x$. If we take the supremum over $x$ on both sides we get

$$\sup_x |\sum_{i=1}^m \sigma'(x^T u_i)\alpha_i - \sum_{i=1}^m \sigma'(x^T u_i')\alpha_i| \leq (\sup_{i,x} l_i^x)|\theta_1 - \theta_2| \tag{191}$$

Denoting $(\sup_{i,x} l_i^x) = l$, we get

$$|\sum_{i=1}^m \sigma'(x^T u_i)\alpha_i - \sum_{i=1}^m \sigma'(x^T u_i')\alpha_i| \leq l|\theta_1 - \theta_2|_1 \tag{192}$$

$$\forall x \in \mathbb{R}^d \tag{193}$$

$$\square$$

Now since the critic step becomes a convex optimization Lemma 6 now becomes

**Lemma 10.** *For a given iteration $k$ of Algorithm 2 and iteration $j$ of the first for loop of Algorithm 2, let the number of steps of the projected gradient descent performed by Algorithm 3, denoted by $T_{k,j}$, and the gradient descent step size $\alpha_{k,j}$ satisfy*

$$\alpha_{k,j} = \frac{||u_{k,j}^*||_2}{L_{k,j}\sqrt{T_{k,j}+1}}, \tag{194}$$

*for some constants $L_{k,j}$ and $||(u_k^*)||_2$. Then the error $\epsilon_{k_4}$ defined in Definition 7 is upper bounded as*

$$\mathbb{E}(|\epsilon_{k,j}^4|) \leq \tilde{\mathcal{O}}\left(\frac{1}{\sqrt{T_{k,j}}}\right) + \epsilon_{|\tilde{D}|}, \tag{195}$$

*Proof.* For a given iteration $k$ of Algorithm 1 and iteration $j$ of the first inner for loop, the optimization problem to be solved in Algorithm 3 is the following

$$\mathcal{L}(\theta) = \frac{1}{n} \sum_{i=1}^{n} \left( Q_\theta(s_i, a_i) - \left( r(s_i, a_i) + \gamma \max_{a' \in \mathcal{A}} \gamma Q_{k,j-1}(s', a') \right) \right)^2 \tag{196}$$

Here, $Q_{k,j-1}$ is the estimate of the $Q$ function from the iteration $j-1$ and the state action pairs $(s_i, a_i)_{i=\{1,\cdots,n\}}$ have been sampled from a distribution over the state action pairs denoted by $\nu$. Since $\min_\theta \mathcal{L}(\theta)$ is a non convex optimization problem we instead solve the equivalent convex problem given by

$$u_{k,j}^* = \arg\min_u g_{k,j}(u) = \arg\min_u || \sum_{D_i \in \tilde{D}} D_i X_{k,j} u_i - y_k ||_2^2 \tag{197}$$

$$subject\ to |u|_1 \leq \frac{R_{\max}}{1-\gamma} \tag{198}$$

Here, $X_{k,j} \in \mathbb{R}^{n \times d}$ is the matrix of sampled state action pairs at iteration $k$, $y_k \in \mathbb{R}^{n \times 1}$ is the vector of target values at iteration $k$. $\tilde{D}$ is the set of diagonal matrices obtained from line 2 of Algorithm 3 and $u \in \mathbb{R}^{|\tilde{D}d| \times 1}$ (Note that we are treating $u$ as a vector here for notational convenience instead of a matrix as was done in Section 4).

The constraint in Equation equation 198 ensures that the all the co-ordinates of the vector $\sum_{D_i \in \tilde{D}} D_i X_{k,j} u_i$ are upper bounded by $\frac{R_{max}}{1-\gamma}$ (since all elements of $X_{k,j}$ are between 0 and 1). This ensures that the corresponding neural network represented by Equation equation 149 is also upper bounded by $\frac{R_{max}}{1-\gamma}$. We use the a projected gradient descent to solve the constrained convex optimization problem which can be written as.

$$u_{k,j}^* = \arg\min_{u:|u|_1 \leq \frac{R_{\max}}{1-\gamma}} g_{k,j}(u) = \arg\min_{u:|u|_1 \leq \frac{R_{\max}}{1-\gamma}} || \sum_{D_i \in \tilde{D}} D_i X_{k,j} u_i - y_k ||_2^2 \tag{199}$$

From Ang, Andersen(2017). "Continuous Optimization" [Notes]. https://angms.science/doc/CVX we have that if the step size $\alpha = \frac{||u_{k,j}^*||_2}{L_{k,j}\sqrt{T_{k,j}+1}}$, after $T_{k,j}$ iterations of the projected gradient descent algorithm we obtain

$$(g_{k,j}(u_{T_{k,j}}) - g_{k,j}(u^*)) \leq L_{k,j} \frac{||u_{k,j}^*||_2}{\sqrt{T_{k,j}+1}} \tag{200}$$

Where $L_{k,j}$ is the lipschitz constant of $g_{k,j}(u)$ and $u_{T_{k,j}}$ is the parameter estimate at step $T_{k,j}$.

Therefore if the number of iteration of the projected gradient descent algorithm $T_{k,j}$ and the step-size $\alpha$ satisfy

$$T_{k,j} \geq L_{k,j}^2 ||u_{k,j}^*||_2^2 \epsilon^{-2} - 1, \tag{201}$$

$$\alpha = \frac{||u_{k,j}^*||_2}{L_{k,j}\sqrt{T_{k,j}+1}}, \tag{202}$$

we have

$$(g_{k,j}(u_{T_{k,j}}) - g_{k,j}(u^*)) \leq \epsilon \tag{203}$$

Let $(v_i^*, w_i^*)_{i \in (1, \cdots, |\tilde{D}|)}$, $(v_i^{T_{k,j}}, w_i^{T_{k,j}})_{i \in (1, \cdots, |\tilde{D}|)}$ be defined as

$$(v_i^*, w_i^*)_{i \in (1, \cdots, |\tilde{D}|)} = \psi_i'(u_i^*)_{i \in (1, \cdots, |\tilde{D}|)} \tag{204}$$

$$(v_i^{T_{k,j}}, w_i^{T_{k,j}})_{i \in (1, \cdots, |\tilde{D}|)} = \psi_i'(u_i^{T_{k,j}})_{i \in (1, \cdots, |\tilde{D}|)} \tag{205}$$

where $\psi'$ is defined in Equation equation 164.

Further, we define $\theta_{|\tilde{D}|}^*$ and $\theta^{T_{k,j}}$ as

$$\theta_{|\tilde{D}|}^* = \psi(v_i^*, w_i^*)_{i \in (1, \cdots, |\tilde{D}|)} \tag{206}$$

$$\theta^{T_{k,j}} = \psi(v_i^{T_{k,j}}, w_i^{T_{k,j}})_{i \in (1, \cdots, |\tilde{D}|)} \tag{207}$$

where $\psi$ is defined in Equation equation 159, $\theta_{|\tilde{D}|}^* = \arg\min_\theta \mathcal{L}_{|\tilde{D}|}(\theta)$ for $\mathcal{L}_{|\tilde{D}|}(\theta)$ defined in Appendix F.

Since $(g(u_{T_{k,j}}) - g(u^*)) \leq \epsilon$, then by Lemma 8, we have

$$\mathcal{L}_{|\tilde{D}|}(\theta^{T_{k,j}}) - \mathcal{L}_{|\tilde{D}|}(\theta_{|\tilde{D}|}^*) \leq \epsilon \tag{208}$$

Note that $\mathcal{L}_{|\tilde{D}|}(\theta^{T_{k,j}}) - \mathcal{L}_{|\tilde{D}|}(\theta_{|\tilde{D}|}^*)$ is a constant value. Thus we can always find constant $C_{k,j}'$ such that

$$C_k' |\theta^{T_{k,j}} - \theta_{|\tilde{D}|}^*|_1 \leq \mathcal{L}_{|\tilde{D}|}(\theta^{T_{k,j}}) - \mathcal{L}_{|\tilde{D}|}(\theta_{|\tilde{D}|}^*) \tag{209}$$

$$|\theta^{T_{k,j}} - \theta_{|\tilde{D}|}^*|_1 \leq \frac{\mathcal{L}(\theta^{T_{k,j}}) - \mathcal{L}(\theta^*)}{C_k'} \tag{210}$$

Therefore if we have

$$T_{k,j} \geq L_{k,j}^2 ||u_{k,j}^*||_2^2 \epsilon^{-2} - 1, \tag{211}$$

$$\alpha_{k,j} = \frac{||u_k^*||_2}{L_{k,j}\sqrt{T_{k,j} + 1}}, \tag{212}$$

then we have

$$|\theta^{T_{k,j}} - \theta^*|_1 \leq \frac{\epsilon}{C_k'} \tag{213}$$

which according to Equation equation 210 implies that

$$C_k' |\theta^{T_{k,j}} - \theta_{|\tilde{D}|}^*|_1 \leq \mathcal{L}_{|\tilde{D}|}(\theta^{T_{k,j}}) - \mathcal{L}_{|\tilde{D}|}(\theta_{|\tilde{D}|}^*) \leq \epsilon \tag{214}$$

Dividing Equation equation 214 by $C_k'$ we get

$$|\theta^{T_{k,j}} - \theta_{|\tilde{D}|}^*|_1 \leq \frac{\mathcal{L}_{|\tilde{D}|}(\theta^{T_{k,j}}) - \mathcal{L}_{|\tilde{D}|}(\theta_{|\tilde{D}|}^*)}{C_k'} \leq \frac{\epsilon}{C_k'} \tag{215}$$

Which implies

$$|\theta^{T_{k,j}} - \theta_{|\tilde{D}|}^*|_1 \leq \frac{\epsilon}{C_k'} \tag{216}$$

Assuming $\epsilon$ is small enough such that $\frac{\epsilon}{C'_k} < 1$ from lemma 9, this implies that there exists an $L_{k,j} > 0$ such that

$$
\begin{align}
|Q_{\theta^{T_{k,j}}}(s,a) - Q_{\theta^*_{|\tilde{D}|}}(s,a)| &\leq L_{k,j}|\theta^{T_{k,j}} - \theta^*_{|\tilde{D}|}|_1 \tag{217} \\
&\leq \frac{L_{k,j}\epsilon}{C'_k} \tag{218}
\end{align}
$$

for all $(s,a) \in \mathcal{S} \times \mathcal{A}$. Equation equation 218 implies that if

$$
\begin{align}
T_{k,j} &\geq L_{k,j}^2 ||u^*_{k,j}||_2^2 \epsilon^{-2} - 1, \tag{219} \\
\alpha_{k,j} &= \frac{||u^*_k||_2}{L_{k,j}\sqrt{T_{k,j}+1}}, \tag{220}
\end{align}
$$

then we have

$$
\mathbb{E}(|Q_{\theta^{T_{k,j}}}(s,a) - Q_{\theta^*_{|\tilde{D}|}}(s,a)|) \leq \frac{L_{k,j}\epsilon}{C'_k} \tag{221}
$$

By definition in section C $Q_{k,j}$ is our estimate of the $Q$ function at the $k^{th}$ iteration of Algorithm 1 and thus we have $Q_{\theta^{T_{k,j}}} = Q_{k,j}$ which implies that

$$
\mathbb{E}(|Q_{k,j}(s,a) - Q_{\theta^*_{\tilde{D}}}(s,a)|) \leq \frac{L_{k,j}\epsilon}{C'_k} \tag{222}
$$

If we replace $\epsilon$ by $\frac{C'_{k,j}\epsilon}{L_{k,j}}$ in Equation equation 221, we get that if

$$
\begin{align}
T_{k,j} &\geq \left(\frac{C'_{k,j}\epsilon}{L_{k,j}}\right)^{-2} L_{k,j}^2 ||u^*_{k,j}||_2^2 - 1, \tag{223} \\
\alpha_{k,j} &= \frac{||u^*_k||_2}{L_{k,j}\sqrt{T_{k,j}+1}}, \tag{224}
\end{align}
$$

we have

$$
\mathbb{E}(|Q_{k,j}(s,a) - Q_{\theta^*_{\tilde{D}}}(s,a)|) \leq \epsilon \tag{225}
$$

From Assumption 5, we have that

$$
\mathbb{E}(|Q_{\theta^*}(s,a) - Q_{\theta^*_{\tilde{D}}}(s,a)|) \leq \epsilon_{|\tilde{D}|} \tag{226}
$$

where $\theta^* = \arg\min_{\theta \in \Theta} \mathcal{L}(\theta)$ and by definition of $Q^3_{k,j}$ in Definition 6, we have that $Q^3_{k,j} = Q_{\theta^*}$. Therefore if we have

$$
\begin{align}
T_{k,j} &\geq \left(\frac{C'_{k,j}\epsilon}{L_{k,j}}\right)^{-2} L_{k,j}^2 ||u^*_{k,j}||_2^2 - 1, \tag{227} \\
\alpha_{k,j} &= \frac{||u^*_k||_2}{L_{k,j}\sqrt{T_{k,j}+1}}, \tag{228}
\end{align}
$$

we have

$$\mathbb{E}(|Q_{k,j}(s,a) - Q_{k,j}^3(s,a)|)_\nu \quad \leq \quad \mathbb{E}(|Q_{k,j}(s,a) - Q_{\theta_{\tilde{D}}^*}(s,a)|) + \mathbb{E}(|Q_{k,j}^3(s,a) - Q_{\theta_{\tilde{D}}^*}(s,a)|) \tag{229}$$

$$\leq \quad \epsilon + \epsilon_{|\tilde{D}|} \tag{230}$$

This implies

$$\mathbb{E}(|Q_{k,j}(s,a) - Q_{k,j}^3(s,a)|) \quad \leq \quad \tilde{\mathcal{O}}\left(\frac{1}{\sqrt{T_{k,j}}}\right) + \epsilon_{|\tilde{D}|} \tag{231}$$

$\square$

Thus replacing lemma 6 with lemma 10 we obtain the following.

**Theorem 2.** *Suppose Assumptions 1-5 hold and we have,* $\alpha_i = \frac{||u_{k,j}^*||_2}{L_{k,j}\sqrt{i+1}}$, $\eta = \frac{1}{\sqrt{K}}$ *and* $\beta_i = \frac{2}{\mu_k(i+1)}$, *then we obtain*

$$\min_{k \leq K}(V^*(\nu) - V^{\pi_{\lambda_K}}(\nu)) \leq \mathcal{O}\left(\frac{1}{\sqrt{K}(1-\gamma)}\right) + \frac{1}{K(1-\gamma)}\sum_{k=1}^{K}\sum_{j=0}^{J-1}\mathcal{O}\left(\frac{\log\log(n)}{\sqrt{n}}\right) +$$

$$+ \frac{1}{K(1-\gamma)}\sum_{k=1}^{K}\sum_{j=0}^{J-1}\mathcal{O}\left(\frac{1}{\sqrt{T_{k,j}}}\right) + \frac{1}{K(1-\gamma)}\sum_{k=1}^{K}\mathcal{O}(\gamma^J)$$

$$+ \frac{1}{1-\gamma}\left(\epsilon_{bias} + (\sqrt{\epsilon_{approx}}) + \epsilon_{|\tilde{D}|}\right) \tag{232}$$

Hence, for $K = \mathcal{O}(\epsilon^{-2}(1-\gamma)^{-2})$, $J = \mathcal{O}\left(\log\left(\frac{1}{\epsilon}\right)\right)$, $n = \tilde{\mathcal{O}}\left(\epsilon^{-2}(1-\gamma)^{-2}\right)$, $T_{k,j} = \mathcal{O}(\epsilon^{-2}(1-\gamma)^{-2})$ we have

$$\min_{k \leq K}(V^*(\nu) - V^{\pi_{\lambda_K}}(\nu)) \leq \epsilon + \frac{1}{1-\gamma}\left(\epsilon_{bias} + (\sqrt{\epsilon_{approx}}) + \epsilon_{|\tilde{D}|}\right), \tag{233}$$

which implies a sample complexity of $\sum_{k=1}^{K}\sum_{j=1}^{J}(n) = \tilde{\mathcal{O}}\left(\epsilon^{-4}(1-\gamma)^{-4}\right)$.

Note that we no longer have the probability statement of 'with a probability at least $1 - \delta$. However, on the flip side, we have and extra error term in the form of $\epsilon_{|\tilde{D}|}$. This error represents the error incurred due to the inability to obtain an exact convex reformulation of the 2 layer neural network.