# OpenReview forum: "On the Global Convergence of Natural Actor-Critic with Neural Network Parametrization"
_ICLR.cc/2024/Conference — Submitted to ICLR 2024_

### Official Review · Reviewer_koRw · 2023-10-20

**Soundness:** 2 fair
**Presentation:** 2 fair
**Contribution:** 2 fair
**Rating:** 3
**Confidence:** 3

**Summary:**

The paper provides a theoretical convergence analysis for natural actor-critic with neural network function approximation for the critic and softmax parameterization for the actor. The authors are able to prove global convergence is proven.

**Strengths:**

This paper studies a problem closely related to the practice. Data are allowed to be Markovian in the analysis, which uses a result from Bertail & Portier (2019), which might be of independent interest.

**Weaknesses:**

First of all, the authors claim in the introduction that existing works that study actor-critic with neural network function approximation only provide asymptotic results. However, I found the following paper, which is not compared with in this paper:

Cayci, Semih, Niao He, and R. Srikant. "Finite-time analysis of entropy-regularized neural natural actor-critic algorithm." arXiv preprint arXiv:2206.00833 (2022).

Could the authors kindly clarify how the result in this submission differentiates from the one in this paper from 2022?

In addition, the authors consider the flexibility of allowing non-i.i.d. samples in their analysis as a contribution of this paper, but the following paper, which the authors have already cited, also allows Markovian data. Could the authors shed some light on how the analysis technique used in this paper differs from the one in the following paper? (I understand their result is only for linear function approximation. Does this make a big difference in the part of analysis pertaining to the concentration over non-i.i.d. Markovian data?)

Xu, Tengyu, Zhe Wang, and Yingbin Liang. "Improving sample complexity bounds for (natural) actor-critic algorithms." Advances in Neural Information Processing Systems 33 (2020): 4358-4369.

The above two questions is mainly concerned with the significance/novelty of this submission. I also have several technical questions, which I will defer to the Questions section.

**Questions:**

- What is the architecture of your neural network in Algorithm 1? How does this function class look like? You stated $L$ layers and $m$ neurons per layer in Section 4, but is there any other requirements or specifications? A paragraph that states this more rigorously would be appreciated.

- A related question is: how does the size of your neural network function class or $\Theta$ affect the variance portion of your final error bound? Your current bound does not show any dependence on it. Could you explain why the upper bound of the expectation of a supremum over $\Theta$ does not need to depend on the complexity of $\Theta$ in Equation (110)? Otherwise, if the Rad quantity in (110) actually depends on $\Theta$, could you explicitly present in your theorem how $C_k$ in your Equation (111) depends on $\Theta$, as most existing theoretical results do. This is an important aspect of such theoretical results, since taking a covering with a supremum usually compromises the sharpness of the bound. If you'd like to compare with the rates in other theoretical works, it might come across as unfair if your bound does not show this aspect.

- Is there any difference between Algorithm 1 and the procedure you actually use in your theoretical analysis? Algorithm 1 seems to be a practical implementation which uses gradient descent to solve the objective in Equation (11), whereas I believe you assume there is an oracle that can find the global minimizer of (11) in your theoretical analysis. (Otherwise, could you explain how you avoided the non-convexity?) If this is the case, this difference should be made much clearer in the writing, and Theorem 1 should not be claimed as a result for the current Algorithm 1.

- Please let me know if the following is correct: $\alpha m$ can be $\Theta(\frac{1}{\mathrm{poly}(n,L)})$, so the second line of Equation (17) can be viewed as $\frac{1}{K(1-\gamma)}\sum_{k=1}^K\sum_{j=0}^{J-1}(1 - \frac{1}{n^2})^n$. Since $(1 - \frac{1}{n^2})^n$ is monotonically increasing and converges to a constant, is this term on the order of $\frac{J}{1-\gamma}$? If this is the case, it is non-decaying.

- What is $\mu_k$ in Theorem 1? Isn't the loss function in Equation (9) just MSE?

- Could you also explain the following in Remark 3:

> This term is present due to the fact that the optimization for the critic step is non-convex, hence convergence can only be guaranteed with a high probability. We show in the Appendix F that if the critic is represented by a two layer neural network with ReLU activation, using the convex reformulation as laid out in Mishkin et al. (2022), a deterministic upper bound on the error can be obtained.

Normally, the high probability bound is due to the randomness in your samples. Could you explain why you attribute it to the non-convexity? Furthermore, when everything is convex, could you explain why you can obtain a deterministic result, when the samples from Line 5 and 10 of Algorithm 1 are still random?

- Also, the notation $n.J$ in Algorithm 1 and Theorem 1 seems weird. Did you mean to write $n\cdot J$? Same with other instances of dots.

Despite my serious concerns at the moment, I'm open to raising my score if authors' clarification could clear my questions.

---

> ### Author Response · Authors · 2023-11-14
> **Reply To Reviewer**
>
> __Weaknesses__
>
> * With regards to the paper Cayci et.al (2022), thank you for pointing it out. Since the paper is on archive, it did not come across in our literature review. However, having gone through it, we can see that in this paper, the actor and critic are both restricted to a two layer neural network, this is in contrast to our work which allows for a multi layer actor and critic. Additionally, note that this work achieves a sample complexity of $\epsilon^{-6}$ which is worse than our result or $\epsilon^{-4}$ even in this restricted setting. We will add this paper in our related works.
>
> *  Secondly with respect to the the non-i.i.d analysis in Xu et al.(2020), the ergodicity assumption is used to bound the difference between the estimate of the advantage function obtained form the Markovian data and that which would have been obtained from i.i.d data. This can be seen in Page 23 equation 25. For our case we account for the Markovian dependence in the critic samples by splitting the critic optimization as given in Equation (25), here the corresponding term is $\epsilon^{3}_{k,j}$. This term is bounded using the theory in Bertail and Porter (2019). Additionally, since we calculate the natural gradient update through the optimization given in equation (29), we account for the Markovian dependence of its samples through the result of Doan (2022). Note that by using this optimization we avoid having to invert our estimate of the fisher information matrix as is the case for Xu et al.(2020).
>
> __Questions__
> * With regards to the architecture of the network, the results we have used from Zhu (2019), were proven for  a fully connected feed-forward neural network with ReLU activation layers. This result also holds for a smooth activation function (the author states that the results can be made sharper for a smooth activation function), this result would also apply for a conventional neural network as well as a residual neural network. For our case we can assume the case of a fully connected neural network with $L$ layers, at least $m$ neurons per layer and with an ReLU activation function for each layer.
>
> * With regards to Equation (110), the set $\Theta$ here does not represent the class of all possible neural networks but only the two neural networks corresponding to the parameters $\theta_{k,j}^2$  and $\theta_{k,j}^3$, this is a typo on our part, we will rename it to $\Theta^{'}$ and state that it is the set of only two parameters $\theta_{k,j}^2$ and $\theta_{k,j}^3$. This is because we only need this bound for the two mentioned parameters as shown in Equation (112) and (113). We will make that change in the final version. With regards to the property of the function class, Bertail and Porter mentions that the function class has to be uniformly bounded. This condition is satisfies for the set $\Theta^{'}$ as it has only two elements and the state action pairs are defined on a bounded space.
>
> * Thank you for pointing out this result, this was an over-site on our part. The term that you mentioned $\left(1-\frac{1}{n^{2}}\right)^{n}$ does indeed converge to $1$ and hence the term on the right you have pointed out will be non-decaying. However, we can resolve this by a minor modification. In lines  $7-11$ of Algorithm 1 we perform stochastic gradient descent with data sample of size $n$ and also putting the number of iterations equal to $n$. We did not consider the edge case which you have mentioned where the error term becomes non-decaying. However this can be remedied with a minor modification. Instead of lines  $7-11$ of Algorithm 1 we will have the following.
>   * Sample and store $n$ state action pairs and the resulting rewards using policy $\pi^{\lambda_{k}}$
>   * Then randomly sample the tuple $(s_{i},a_{i},r(s_{i},a_{i}),s_{i+1})$ from the stored data, calculate $y_{i}$ and preform the gradient descent step as in line 9 and 10 of Algorithm 1 respectively. This random sampling and update will be performed for $T_{k,j}$ iterations.
>
> This will result in the superscript in the term to be $T_{k,j}$ instead of $n$. This can be seen from Equation 13.2 on page 34 of Zhu (2019) (please note that for some reason, the supplementary part of this paper is not available on the PMLR website, but is available on the archive version). Now using the result of Theorem 2 of  Zhu (2019), for $T_{k,j} = \Theta\left(poly(n,L)\cdot{\log\frac{1}{\epsilon}}\right)$ the term inside the summation will be upper bounded by epsilon.
>
> * The term $\mu_{k}$ represents the strong convexity parameter in Equation (9), since the function in Equation (9) is the expectation of a square loss function, we have to consider its strong convexity parameter.

---

> > ### Author Response · Authors · 2023-11-14
> > **Continuation of Reply**
> >
> > * The high probability term is due to the random initialization of the parameter estimates. (Mishkin 2022) showed that the optimization for a 2 layer neural network with ReLU activation function can be transformed into a convex problem. For this optimization the parameters are initialized. We did not use Algorithm 1 for this case. We used Algorithm 2 (the optimization is carried out in Algorithm 3 where in line 3 the parameters are initialised to zero) in Appendix E.
> > * We apologize for the error. We will correct the notation and use  $\cdot$ instead of .
> >
> > __Bibliograpahy__
> >
> > Cayci, Semih, Niao He, and R. Srikant. "Finite-time analysis of entropy-regularized neural natural actor-critic algorithm." arXiv preprint arXiv:2206.00833 (2022).
> >
> > Wu, Yue Frank, et al. "A finite-time analysis of two time-scale actor-critic methods." Advances in Neural Information Processing Systems 33 (2020): 17617-17628.
> >
> > Bertail, Patrice, and François Portier. "Rademacher complexity for Markov chains: Applications to kernel smoothing and Metropolis–Hastings." (2019): 3912-3938.
> >
> > Doan, Thinh T. "Finite-time analysis of markov gradient descent." IEEE Transactions on Automatic Control 68.4 (2022): 2140-2153.
> >
> > Allen-Zhu, Zeyuan, Yuanzhi Li, and Zhao Song. "A convergence theory for deep learning via over-parameterization." International conference on machine learning. PMLR, 2019.

---

> > > ### Author Response · Authors · 2023-11-19
> > >
> > > Dear Reviewer,
> > >
> > > Thank you so much for your time and efforts in reviewing our paper. We have addressed your comments in detail and are happy to discuss more if there are any additional concerns. We are looking forward to your feedback and would greatly appreciate you consider raising the scores.
> > >
> > > Thank you,
> > >
> > > Authors

---

### Official Review · Reviewer_vh1k · 2023-10-24

**Soundness:** 3 good
**Presentation:** 2 fair
**Contribution:** 3 good
**Rating:** 6
**Confidence:** 5

**Summary:**

The authors propose an actor-critic method with neural network parametrization to solve the RL problem with discrete time, a finite action space and a (potentially) continuous state space. A natural policy gradient method is used for the actor and a method similar to the deep Q-net is applied for the critic. The authors give a global convergence result for the algorithm.

**Strengths:**

The authors propose a new method to solve the RL problem, which has a nice convergence result. I would suggest the paper be published if the author could address my concerns and problems.

**Weaknesses:**

The presentation is not clear enough and there are many mistakes. The correctness of the main theorem remains questionable. I will put the details in “Questions”.
For example, the authors did not give definition for the notation r’(s,a), which caused some confusion. I will, according to the description of Definition 2, understand r’(s,a) as a realization of R(\cdot | s,a), which is the same as r(s,a).

**Questions:**

Major questions:
1.	Page 3. The definition of Bellman operator. In (3), the authors use one realization of the reward r’(s,a) to define the Bellman operator. This makes (T^\pi Q)(s,a) a random variable. So, T^\pi Q^\pi = Q^\pi does not hold. Maybe the authors can consider the setting when the reward is deterministic, otherwise there is a lot to fix. A more challenging alternative is to redefine the Bellman operator with r’(s,a) replaced by its conditional expectation. As a follow up, when T^\pi Q is not deterministic, \eps_{approx} in Assumption 4 could be as large as O(1) according to the variance of the reward. In this case, Theorem 1 becomes meaningless. As another follow up, in page 8 eqn (24), Lemma 2 of Munos (2003) is also for deterministic reward.
2.	Page 4 before (8). The authors use the property of compatible function approximation, which is an extremely strong assumption for neural network parametrization. According to Sutton, this means \frac{\partial Q}{\partial \theta} = \frac{\partial \pi}{\partial \lambda} / \pi. And Sutton also points out (in the paper that the author cited) that “Q being linear in the features may be the only way to satisfy this condition”. On the one hand, this assumption is too strong to make the work practical. On the other hand, there is no critic parametrization until eqn (10). So, I think a normal version of the policy gradient theorem (theorem 1 in the ref) is enough.
3.	Theorem 1 (and also Lemma 7). The authors give no description or assumption on \mu_k. Therefore, the constants for big O in the theorem depend on \mu_k, and hence depend on K. This makes the theorem meaningless. I would suggest giving a positive lower bound for \mu_k. Please note that \mu_k is also the lower bound for the (eigenvalue of the) Fisher information matrix (7), which guarantees that F is invertible.
4.	Lemma 6. According to the description of Lemma 6, there is no randomness in (50) and I don’t see why “with high probability” is needed. My understanding is that (50) left is not an expectation, but a conditional expectation only w.r.t. the RL problem. The randomness comes from the initialization of neural network etc. However, the authors give no description of the network (except the width). I believe inequality like (50) could only hold when one gives a very explicit setting for everything. Please clarify the setting with details so that we can justify the application of the referenced theorems.
5.	Proof of theorem 1. Eqn (56) is not consistent with line 20 in the algorithm. There should be a coefficient 1/(1-\gamma). I think the statement of theorem 1 need modification accordingly, because you are tracking the dependence on 1/(1-\gamma). (58)-(60), when you define err_k, do you mean to use A^{\pi_{\labmda_k}} instead of A_{k,J}? Otherwise, it is not consistent with (66).

Minor questions:
1.	Page 2. The sentence “the non-convexity of the critic can be avoided if we use the natural policy gradient algorithm” is not clear. What do you mean by “the non-convexity of the critic”? Natural policy gradient is the actor algorithm, why does it resolve the problem for the critic?
2.	Page 2 Related works: actor-critic methods. I think papers like “Provably global convergence of actor-critic: A case for linear quadratic regulator with ergodic cost” (NIPS2019) and “Single Timescale Actor-Critic Method to Solve the Linear Quadratic Regulator with Convergence Guarantees” (JMLR2023) are also related.
3.	Page 3. Why does the stationary distribution \rho^\pi_\nu depend on the initial distribution \nu, especially when Assumption 2 holds?
4.	Page 4. The definition of natural policy gradient method. I believe that the dagger means inverse of pseudo-inverse of the matrix. It should not appear in eqn (7), where the Fisher information matrix is defined. Also, please clarify the dagger notation.
5.	Page 4. Please give the full name of DQN when it first appears.
6.	Page 6 after (15). In Xu and Gu (2020), I did not find the assumption described in the paper.
7.	Proof of theorem 1. (53)(54) looks unnecessary when you have (55)(56). (54) looks wrong. (58) should be >= instead of <=? Eqn (58) to (59) has nothing to do with the performance difference lemma, maybe you mean eqn (59) to (60)? (59) first line, should use the advantage function instead of the Q function? (59) second line, should be + instead of -? Eqn (63), why does k goes to J-1 instead of K? Also, in (63), \pi_{\lambda_0} should be \pi_{\lambda_1}? It seems that one of \eta is omitted from (62) to (63), which is not wrong but makes the proof harder, why doing this?
8.	Please add an explanation for the log(|A|) in eqn (64).

There also some Typos:
Page 3 after “Hence, we can write” d^\pi(s_0) should be d^\pi_{s_0}(s)
Page 4. \Lambda should be a subset (not element) of R^d
Page 5 Algorithm. Line 12, should be Q_k = Q_{\theta_n}?
Page 6 eqn (15). w^t should be w?
In theorem 1, Lemma 6,7, many of the dots (as product) are written as periods.
Theorem 1 eqn (17)-(19) and in the proof. \lambda_K should be \lambda_k?
Page 9 Upper Bounding Error in Actor Step. There should be a square in the expression below.
Page 14 Def 9. Z is a subset of R^n.
Lemma 2 consider three random variables.
Page 16 eqn (47) bias should be approx.
Page 16 bottom, \theta_k should be \lambda_k, argmin should be min.
Page 17 eqn (57)-(62), lack the second “KL(”.

---

> ### Author Response · Authors · 2023-11-14
> **Reply To Reviewer**
>
> __Major Questions__
> * We apologise for the error in the definition of $T^{\pi}Q(s,a)$. It should have been defined as $T^{\pi}Q^{\pi}(s,a) = \mathbb{E}(r(s,a)) + {\gamma}{\int}Q(s^{'},\pi(s^{'}))P(ds^{'}|s,a)$, where $\mathbb{E}(r(s,a))$ is the expected reward given the state action pair $(s,a)$. This would make $T^{\pi}Q(s,a)$ a fixed value and not a random variable. With regards to our comment about about the result in Munos (2003), we referenced that result as a result  similar to that was used in Faharmand (2010), which we used for our analysis. As for the result we used (in Equations 74-77), it depends upon the property $T^{\pi}Q^{\pi}(s,a)=Q^{\pi}(s,a)$. This property will hold even for a random reward function. This is shown as follows.
>
>     \begin{eqnarray}
>         Q^{\pi}(s,a) &=& \mathbb{E}\left[ \sum_{t=0}^{\infty}{\gamma}^{t}r(s_{t},a_{t})|s_{0}=s,a_{0}=a\right] \\\\
>                     &=& \mathbb{E}(r(s,a))  + \gamma \mathbb{E} \left[\sum_{t=1}^{\infty} {\gamma}^{t-1}r(s_{1},a_{1})|s_{0}=s,a_{0}=a \right] \\\\
>                      &=&  \mathbb{E}(r(s,a))  + {\gamma}{\int}Q(s^{'},\pi(s^{'}))P(ds^{'}|s,a) \\\\
>                      &=&  T^{\pi}Q^{\pi}
>     \end{eqnarray}
>
>
>     We will define $r^{'}(s,a)$ as the realisation of the random reward at the state action pair. We will also replace $r^{'}(s,a)$ in Equation (27) and replace $r(s,a)$ with $r^{'}(s,a)$ in Equation (28). Additionally, in Algorithm 1, $r(s,a)$ will be replaced by  $r^{'}(s,a)$.
>
> * With regards to the reference of Sutton (1999), we would like to point out that before it states that a linear parameterization of the critic is required, in the line just above this statement (at the bottom of page 3), the  author assumes that the policy is being parameterised using a linear function. It is for this reason that a linear parametrization of the critic is required. In our case where we assume that the policy parametrization is smooth and not necessarily linear, restricting the critic to be a linear function would also require us to restrict the policy class to be linear. This is not something that is done typically in practice. As for the piratical feasibility of using a neural network for the critic, it is shown in Bharadwaj (2022), natural actor critic with a neural parameterization does converge.
>
> * Thank you for pointing this out. We agree that we should have added an additional assumption in order to ensure the constant appearing on the loss function is independent ok $K$. Thus the additional assumption will be as follows.
>     * There exists a constant $\\mu$ such that for any policy $\pi_{\lambda}, \lambda \in \Lambda$ we have
>
>         \begin{eqnarray}
>          \mathbb{E}_{s,a \sim \zeta^{\lambda}} \left(\nabla{\log}(\pi(a|s))\nabla{\log}(\pi(a|s))^{T}\right)  \succeq
>          \mu{\cdot}I_d
>          \end{eqnarray}
>
>         Where  $\zeta^{\lambda}$ is the stationary state distribution corresponding to the policy $\pi_{\lambda}$, $I_d$ is the identity matrix of dimension $d$ which is the dimensionality of  $\Lambda$.
>
> * The randomness in Equation (50) is as you suggested because of the random initialization of the neural network parameters during the critic step. With regards to the architecture of the network, the results we have used from Zhu (2019), which  were proven for a fully connected feed-forward neural network with ReLU activation layers. The results also holds for a smooth activation function (the author states that the results can be made sharper for a smooth activation function). The result would also apply for a conventional neural network as well as a residual neural network. For our case we can assume the case of a fully connected neural network with $L$ layers, at least $m$ neurons per layer and with an ReLU activation function for each layer.
>
> *  Thank you for pointing this out. We need to change the natural policy update is given by $\lambda_{k+1} = \lambda_{k} + {\eta}w_{k}$. This is in keeping with the analysis in Agarwal, (2020). We explain in Appendix $E$ how our analysis extends upon the work done for the natural policy gradient to that of a natural actor critic.

---

> ### Author Response · Authors · 2023-11-14
> **Continiuation of Reply**
>
> __Minor Questions__
>
> * When we are implementing the natural actor critic, if the critic parameterization is non-convex (as is the case for a neural network parametrization) the estimation of the critic at an iteration of the natural actor critic algorithm becomes a non-convex problem. This is the reason why prior works such as (Xu et al. 2020) used a linear parameterization of the critic. Natural Policy Gradient algorithms do not have this issue as they take a sample based estimate of the critic at each iteration. The reason that they avoid this problem is because they do not account for the error incurred due to sample estimate of the critic.
> * Thank you for pointing out these works. We will incorporate them in the related works, the reason we did not focus on the case of Linear Quadratic regulators is because we were focusing on existing literature where linear critic is used. A linear quadratic regulator would be a further simplification and hence not our focus.
> *  They stationary distribution as you have pointed out does indeed not depend upon the starting distribution if the Markov chain is ergodic, we will make the required change.  However, we do not know of any rigorous proof to show that the visitation distribution will not depend upon the initial distribution.
>  * Yes the dagger notation should not be present in Equation (7), thank you for pointing this out. We will add that the dagger notation represents the pseudo inverse.
>  * We will rename it to Deep Q Networks algorithm.
> *  This is a typo. The paper were referring to here was Liu et al. (2020). The assumption we referred to can be seen in Assumption 4.4 on page 7.
> *  Thank you pointing out these mistakes. We will correct them in the final version. We will also add discussions about the $|\log(\mathcal{A})|$ term that you mentioned.
>
>
> __Bibliograpahy__
>
> Munos, Rémi. "Error bounds for approximate policy iteration." ICML. Vol. 3. 2003.
>
> Farahmand, Amir-massoud, Csaba Szepesvári, and Rémi Munos. "Error propagation for approximate policy and value iteration." Advances in Neural Information Processing Systems 23 (2010)
>
> Sutton, Richard S., et al. "Policy gradient methods for reinforcement learning with function approximation." Advances in neural information processing systems 12 (1999).
>
> Diddigi, Raghuram Bharadwaj, Prateek Jain, and Shalabh Bhatnagar. "Neural network compatible off-policy natural actor-critic algorithm." 2022 International Joint Conference on Neural Networks (IJCNN). IEEE, 2022.
>
> Allen-Zhu, Zeyuan, Yuanzhi Li, and Zhao Song. "A convergence theory for deep learning via over-parameterization." International conference on machine learning. PMLR, 2019
>
> Agarwal, Alekh, et al. "On the theory of policy gradient methods: Optimality, approximation, and distribution shift." The Journal of Machine Learning Research 22.1 (2021)
>
> Xu, Tengyu, Zhe Wang, and Yingbin Liang. "Improving sample complexity bounds for (natural) actor-critic algorithms." Advances in Neural Information Processing Systems 33 (2020).
>
> Liu, Yanli, et al. "An improved analysis of (variance-reduced) policy gradient and natural policy gradient methods." Advances in Neural Information Processing Systems 33 (2020).

---

> > ### Author Response · Authors · 2023-11-19
> >
> > Dear Reviewer,
> >
> > Thank you so much for your time and efforts in reviewing our paper. We have addressed your comments in detail and are happy to discuss more if there are any additional concerns. We are looking forward to your feedback and would greatly appreciate you consider raising the scores.
> >
> > Thank you,
> >
> > Authors

---

> > > ### Comment · Reviewer_vh1k · 2023-11-22
> > > **The revision looks good and I decide to raise my assesment to 6**
> > >
> > > The revision looks good and I decide to raise my assesment to 6

---

### Official Review · Reviewer_x5yz · 2023-10-30

**Soundness:** 3 good
**Presentation:** 3 good
**Contribution:** 2 fair
**Rating:** 5
**Confidence:** 5

**Summary:**

This paper studies the convergence of NAC under two two-layer NN settings. The authors gives a $\epsilon^{-4}(1 - \gamma)^{-4}$ sample complexity for the guarantee for global convergence.

**Strengths:**

- This paper provides a non-asymptotic sample complexity for NAC under two-layer NN, compared to previous work, the non-asymptotic bound is more challenging.
- This paper is well-written and easy to follow.

**Weaknesses:**

- The contribution of this paper needs to be highlighted: it seems that the result can be provided by combining the NTK analysis and the global convergence of NAC (Xu et al., 2020a). Since NTK analysis views the neural networks as a kernel method, the analysis is just let the width of $m$ be very large and then does the error analysis given the linear function.
- Given the aforementioned issue, the result $\epsilon^{-4}(1 - \gamma)^{-4}$ cannot match the previous result $\epsilon^{-3}(1 - \gamma)^{-4}$ in the linear case. I suspect the addition $\epsilon^{-1}$ is sacrificed for the neural network approximation errors

Therefore, I would encourage the authors to highlight the contribution of this paper based on the current well-developed NTK theory and  NAC theory.

**Questions:**

Please see the weakness.

---

> ### Author Response · Authors · 2023-11-14
> **Reply To Reviewer**
>
> __Weaknesses__
>
> * With regards to your comment about combining NTK analysis with  for the work of Xu et al. (2020a), we would like to  __point out a key deficiency in that approach__. In Xu et al. (2020a), while the policy class is non linear, the critic class has been restricted to a linear function. Now note that in Sutton(1999) on page 5, it is stated that a critic is restricted to be linear only when the policy class is also parameterized by a linear function for the compatible function approximation to hold. Thus in order for the compatible function approximation to hold, in the limit of the of the infinite width, both the actor and critic should converge to a linear kernel. If such a result is achieved, we will not recover the result of  Xu et al. (2020), where the policy is still non-linear and the critic is linear. As  Xu et al. (2020a) itself notes, the term $\zeta^{actor}_{approx}$ (which is the term Sutton (1999) references) can be said to be small for an  overparameterized neural policy according to Wang (2019). This is mentioned in page 8 just before Theorem 3. However, Wang (2019) states that __both__ actor and critic are represented by a neural network for this error to be small, something that is not possible in  Xu et al. (2020a) as the critic has to be linear.
>
>     __However, results achieving convergence for the natural actor critic in the infinite width limit have been proven and our result improves upon them.__
>
>     Indeed, Wang (2019), mentioned in Xu et al. (2020a) is such a work. It establishes an upper bound only for the case where both the actor and critic are restricted to a two layer network with ReLU activation.
>     If you look at the convergence result in Theorem A.4. We see that while the iteration complexity of the result is $\mathcal{O}(T^{-\frac{1}{2}})$ the sample complexity required for the to achieve an $\epsilon$ error bound is $\epsilon^{-8}$. additionally, we see that the extra error due to neural network architecture, note the $\mathcal{O}(m^{-\frac{1}{8}})$ term in Theorem A.4, means that the minimum width in the neural network required is of the order $\mathcal{O}(\epsilon^{-\frac{1}{8}})$. For our result, we achieve a sample complexity of $\mathcal{O}(\epsilon^{-\frac{1}{4}})$ and the minimum width in the neural network required is of the order $\mathcal{O}(\epsilon^{-\frac{1}{2}})$.
>
>     Similarly for the result in Fu et al. (2020), while the requirement for a two layer actor and critic is extended to a multi layer actor and critic, the sample complexity even in the asymptotic limit is $\mathcal{O}(\epsilon^{-\frac{1}{6}})$ as can be seen in Theorem C.5. Additionally the minimum width in the neural network required is of the order $\mathcal{O}(T^{-\frac{1}{6}})$, where $T$ is the number of iterations of the algorithm. Since the iteration complexity is given by $\mathcal{O}(\epsilon^{-\frac{1}{2}})$, the minimum width in the neural network required is of the order $\mathcal{O}(\epsilon^{-\frac{1}{12}})$. Thus both in terms of sample complexity as well as the minimum width of the neural network, our result improves upon it.
>
>      We would also like to point out that both these works assume that sample can independently sampled from the stationary state distribution of a given policy, something which is not possible in practice. Our work does not have this restriction. Additionally, both prior works require the use of energy based policies, our analysis is more general, which includes energy based policies but is not restricted to them. Finally, we would like to point out that these results also do not have any empirical studies.
>
>  * As per the reviewers suggestion, we will add more references related to NTK theory.
>
> __Bibliography__
>
> Xu, Tengyu, Zhe Wang, and Yingbin Liang. "Improving sample complexity bounds for (natural) actor-critic algorithms." Advances in Neural Information Processing Systems 33 (2020).
>
> Sutton, Richard S., et al. "Policy gradient methods for reinforcement learning with function approximation." Advances in neural information processing systems 12 (1999).
>
> Wang, Lingxiao, et al. "Neural policy gradient methods: Global optimality and rates of convergence."  In International Conference on Learning Representations, 2019.
>
> Fu, Zuyue, Zhuoran Yang, and Zhaoran Wang. "Single-timescale actor-critic provably finds globally optimal policy."   In International Conference on Learning Representations, 2020.
>
> Allen-Zhu, Zeyuan, Yuanzhi Li, and Zhao Song. "A convergence theory for deep learning via over-parameterization." International conference on machine learning. PMLR, 2019.

---

> > ### Author Response · Authors · 2023-11-19
> >
> > Dear Reviewer,
> >
> > Thank you so much for your time and efforts in reviewing our paper. We have addressed your comments in detail and are happy to discuss more if there are any additional concerns. We are looking forward to your feedback and would greatly appreciate you consider raising the scores.
> >
> > Thank you,
> >
> > Authors

---

### Official Review · Reviewer_1Jb4 · 2023-10-30

**Soundness:** 4 excellent
**Presentation:** 3 good
**Contribution:** 2 fair
**Rating:** 3
**Confidence:** 4

**Summary:**

The authors study natural actor-critic with multi-layer neural network as the function approximation, which builds upon their unique decomposition of the error in the critic steps.

**Strengths:**

- The paper extends previous on natural actor critic to the case of multi-layer neural network and studies the sample complexity of the algorithm. The theoretical study is solid.
- The presentation of the paper is great.
- The paper relaxes the assumptions of previous study of natural actor critic. In particular, the paper does not require i.i.d. sampling.

**Weaknesses:**

- The paper does not provide any empirical study on the algorithm and does not discuss on the empirical implicaiton.
- The authors claim that they are the first to show the sample complexity of natural actor critic algorithms with neural networks. However, to my understanding, the asymptotic converge result in Wang et al, 2019 and Fu et al, 2020 can be converted to an upper bound on the sample complexity directly. Given that, the contribution in this study is rather incremental.
- Missing reference:
Agarwal, Alekh, et al. "On the theory of policy gradient methods: Optimality, approximation, and distribution shift." The Journal of Machine Learning Research 22.1 (2021): 4431-4506.

**Questions:**

- In Theorem 1, $J.n$ should be written as $J\cdot n$.
- The paper does not present a detailed description on how to parameterize the Q function and the policy with neural network in the main part of the paper. A detailed description or a hyper-link to it is helpful.
- The analysis and the error rate of the neural network seems to lie in the regime of neural tangent kernel. Papers on neural tangent kernel should be cited properly. Here are a few examples:
  - Jacot, Arthur, Franck Gabriel, and Clément Hongler. "Neural tangent kernel: Convergence and generalization in neural networks." Advances in neural information processing systems 31 (2018).

---

> ### Author Response · Authors · 2023-11-14
> **Reply To Reviewer**
>
> __Weaknesses__
>
> * We remark that our current work is theoretical in nature where we establish the global optimality of already existing natural actor critic algorithm. A rigorous theoretical understand of global convergence was missing from the literature and our paper fills that gap. Regarding the empirical study, we believe that it would not add any additional value to our contributions because it has already been widely shown in the literature that the natural actor-critic algorithm with neural network parameterization works very well in practice as in Bharadwaj (2022) and has real world applications such as Wang (2021). The focus of our work is establishing that theoretically.
>
>
> * With regard to the two papers mentioned, we respectfully disagree with the reviewer.  Let us consider them individually.
>
>     1. First, Wang (2019), establishes an upper bound only for the case where both the actor and critic are restricted to a two-layer network with ReLU activation. Furthermore, if we look at the convergence result for the neural policy gradient in Theorem A.4. We see that while the iteration complexity of the result is $\mathcal{O}(T^{-\frac{1}{2}})$ the sample complexity required for the to achieve an $\epsilon$ error bound is $\epsilon^{-8}$. Additionally, we see that the extra error due to neural network architecture, note the $\mathcal{O}(m^{-\frac{1}{8}})$ term in Theorem A.4, which means that the minimum width in the neural network required is of the order $\mathcal{O}(\epsilon^{-\frac{1}{8}})$. For our result, we achieve a sample complexity of $\mathcal{O}(\epsilon^{-\frac{1}{4}})$ and the minimum width in the neural network required is of the order $\mathcal{O}(\epsilon^{-\frac{1}{2}})$.
>    2. Secondly,  in Fu et al. (2020), while the requirement for a two layer actor and critic is extended to a multi layer actor and critic, the sample complexity even in the asymptotic limit is $\mathcal{O}(\epsilon^{-\frac{1}{6}})$, this can be seen in the discussion below theorem C.5. Also, the minimum width in the neural network required is of the order $\mathcal{O}(T^{-\frac{1}{6}})$, where $T$ is the number of iterations of the algorithm. Since the iteration complexity is given by $\mathcal{O}(\epsilon^{-\frac{1}{2}})$, the minimum width in the neural network required is of the order $\mathcal{O}(\epsilon^{-\frac{1}{12}})$. Thus both in terms of sample complexity as well as the minimum width of the neural network, our result improves upon the results mentioned.
>
> __Other novelties in our work__ We would also like to point out that both these works assume that sample can independently sampled from the stationary state distribution of a given policy, something which is not possible in practice. Our work does not have this restriction. Also, both prior works require the use of energy based policies, our analysis is more general, which includes energy based policies but is not restricted to them. Finally, we would like to point out that these results also do not have any empirical studies.
>
> __Questions__
>
> * Thank you for pointing this out, we will correct this typo.
>
> * With regards to the architecture of the network, the results we have used from Zhu (2019), which  were proven for a fully connected feed-forward neural network with ReLU activation layers. The results also holds for a smooth activation function (the author states that the results can be made sharper for a smooth activation function). The results in Zhu (2019) would also apply for a conventional neural network as well as a residual neural network. For our case we can assume the case of a fully connected neural network with $L$ layers, at least $m$ neurons per layer and with an ReLU activation function for each layer.
>
> * It seems we have cited the version of this paper published in COLT and not the one in JMLR. We will cite the JMLR version in the final version of our paper.
>
> * We will add the stated reference in the related works section of our final paper.
>
> __Bibliography__
>
> Diddigi, Raghuram Bharadwaj, Prateek Jain, and Shalabh Bhatnagar. "Neural network compatible off-policy natural actor-critic algorithm." 2022 International Joint Conference on Neural Networks (IJCNN). IEEE, 2022.
>
> R. Wang, J. Li, K. Wang, X. Liu and X. Lit, "Service Function Chaining in NFV-Enabled Edge Networks with Natural Actor-Critic Deep Reinforcement Learning," 2021 IEEE/CIC International Conference on Communications in China (ICCC), Xiamen, China, 2021
>
> Wang, Lingxiao, et al. "Neural policy gradient methods: Global optimality and rates of convergence."  In International Conference on Learning Representations, 2019.
>
> Fu, Zuyue, Zhuoran Yang, and Zhaoran Wang. "Single-timescale actor-critic provably finds globally optimal policy."   In International Conference on Learning Representations, 2020.
>
> Allen-Zhu, Zeyuan, Yuanzhi Li, and Zhao Song. "A convergence theory for deep learning via over-parameterization." International conference on machine learning. PMLR, 2019.

---

> > ### Author Response · Authors · 2023-11-19
> >
> > Dear Reviewer,
> >
> > Thank you so much for your time and efforts in reviewing our paper. We have addressed your comments in detail and are happy to discuss more if there are any additional concerns. We are looking forward to your feedback and would greatly appreciate you consider raising the scores.
> >
> > Thank you,
> >
> > Authors

---

### Comment · Area_Chair_3Y4C · 2023-11-14

Dear Authors,

Please note the recent paper

https://openreview.net/forum?id=QlfGOVD5PO

which should be included in the comparisons you make. There are obviously important differences between the above and your work (e.g., actor-critic vs natural actor-critic) but given the similarities, some discussion and comparison should be included in your paper.

---

> ### Author Response · Authors · 2023-11-16
> **Reply to Area Chair**
>
> Thank you for bringing this work to our notice. As you said, this work deals with an actor critic algorithm which is slightly different from our natural actor setup. We would also like to point out that the results here have been proven for a finite state space, an assumption that can be violated even for a simple problem such as the cartpole in OpenAI Gym.
> Nonetheless, we will add a discussion of this in our final version as a concurrent work.

---

### Author Response · Authors · 2023-11-22

As the author-reviewer discussion period will end soon, we will appreciate it if you could check our response to your review comments. This way, if you have further questions and comments, we can still reply before the author-reviewer discussion period ends. If our response resolves your concerns, we kindly ask you to consider raising the rating of our work. Thank you very much for your time and efforts!

---

### Meta-Review · Area_Chair_3Y4C · 2023-12-06

**Metareview:**

Most of the reviewers are skeptical about this paper: the main issue seems to be the lack of a major contribution relative to the existing literature, which already contains a number of results for actor critic with neural networks, some with arbitrary depth; in particular, as has been pointed out to the authors, the discussion of the previous work in this manuscript is not complete. The methodologies of all these papers also overlaps. To be accepted, this manuscript needs a sharp statement about what is really novel in this result and what is difficult relative to the existing literature, which is currently missing.

An additional note: as one reviewer points out, the lack of a log (1/delta) term in Theorem 1 is odd, as is the condition m=>delta^{-1}. Any revision of this work should include a comment discussing these.

**Justification For Why Not Higher Score:**

See metareview.

**Justification For Why Not Lower Score:**

N/A

---

### Decision · Program_Chairs · 2024-01-16

Reject